METHODS AND RESOURCES

# Single-cell atlas of the human neonatal small intestine affected by necrotizing enterocolitis

**Adi Egozi**[1‡], **Oluwabunmi Olaloye**[2‡], **Lael Werner**[3,4], **Tatiana Silva**[2], **Blake McCourt**[2], **Richard W. Pierce**[2,5], **Xiaojing An**[6], **Fujing Wang**[6], **Kong Chen**[6], **Jordan S. Pober**[5,7], **Dror Shouval**[3,4], **Shalev Itzkovitz**[1‡]*, **Liza Konnikova**[2,5,7,8‡]*

**1** Department of Molecular Cell Biology, Weizmann Institute of Science, Rehovot, Israel, **2** Department of Pediatrics, Yale School of Medicine, New Haven, Connecticut, United States of America, **3** Institute of Gastroenterology, Nutrition and Liver Diseases, Schneider Children's Medical Center of Israel, Petah Tikva, Israel, affiliated to the Faculty of Medicine, Tel Aviv University, Tel Aviv, Israel, **4** Sackler Faculty of Medicine, Tel Aviv University, Tel Aviv, Israel, **5** Program in Human and Translational Immunology, Yale School of Medicine, New Haven, Connecticut, United States of America, **6** Department of Medicine, University of Pittsburgh Medical Center Montefiore Hospital, Pittsburgh, Pennsylvania, United States of America, **7** Department of Immunobiology, Yale School of Medicine, New Haven, Connecticut, United States of America, **8** Department of Obstetrics, Gynecology and Reproductive Sciences, Yale School of Medicine, New Haven, Connecticut, United States of America

‡ AE and OO share first authorship on this work. SI and LK are joint senior author on this work.
* shalev.itzkovitz@weizmann.ac.il (SI); liza.konnikova@yale.edu (LK)

**Data Availability Statement:** All data generated in this study is available at the Zenodo repository under the following https://doi.org/10.5281/zenodo.5813397.

## Abstract

Necrotizing enterocolitis (NEC) is a gastrointestinal complication of premature infants with high rates of morbidity and mortality. A comprehensive view of the cellular changes and aberrant interactions that underlie NEC is lacking. This study aimed at filling in this gap. We combine single-cell RNA sequencing (scRNAseq), T-cell receptor beta (TCRβ) analysis, bulk transcriptomics, and imaging to characterize cell identities, interactions, and zonal changes in NEC. We find an abundance of proinflammatory macrophages, fibroblasts, endothelial cells as well as T cells that exhibit increased TCRβ clonal expansion. Villus tip epithelial cells are reduced in NEC and the remaining epithelial cells up-regulate proinflammatory genes. We establish a detailed map of aberrant epithelial–mesenchymal–immune interactions that are associated with inflammation in NEC mucosa. Our analyses highlight the cellular dysregulations of NEC-associated intestinal tissue and identify potential targets for biomarker discovery and therapeutics.

## Introduction

Each year in the United States more than half a million infants are born prematurely. Necrotizing enterocolitis (NEC) is a devastating gastrointestinal (GI) complication that is associated with the degree of prematurity and with high rates of mortality and morbidity. NEC most often affects infants born at <32 weeks' gestation and the onset of symptoms occurs 2 to 8 weeks after delivery [1]. The current incidence of NEC is 1% to 7% of all the infants admitted to the Neonatal Intensive Care Unit with prevalence rising up to 15% for the most premature infants [2]. A recent analysis in infants born prior to 29 weeks' gestation showed a decline in

**Funding:** O.O. is supported by Yale University start-up funds, Patterson Mentored Trust Research award and the CTSA Grant Number KL2TR001862 from the National Center for Advancing Translational Science (NCATS), a component of the NIH. L.K. is supported by Yale University start-up funds, Yale Program for the Promotion of Interdisciplinary Science, Binational Science Foundation award number 2019075 and NIH grants R21TR002639, R21HD102565, and R01AI171980. S.I. is supported by the Wolfson Family Charitable Trust, the Edmond de Rothschild Foundations, the Fannie Sherr Fund, the Dr. Beth Rom-Rymer Stem Cell Research Fund, the Helen and Martin Kimmel Institute for Stem Cell Research, a research grant from the Richard F. Goodman Yale/Weizmann Exchange Program, the Minerva Stiftung grant, the Israel Science Foundation grant no. 1486/16, the European Research Council (ERC) under the European Union's Horizon 2020 research and innovation programme grant no. 768956 and the Chan–Zuckerberg Initiative grant no. CZF2019-002434. The funders had no role in study design, data collection and analysis, decision to publish, or preparation of the manuscript.

**Competing interests:** The authors have declared that no competing interests exist.

**Abbreviations:** CMV, cytomegalovirus; CTCF, corrected total cell fluorescence; DC, dendritic cell; DGE, differential gene expression; FDR, false discovery rate; FFPE, formalin-fixed, paraffin embedded; GI, gastrointestinal; GSEA, gene set enrichment analysis; IBD, inflammatory bowel disease; ILC, innate lymphoid cell; IMC, imaging mass cytometry; NEC, necrotizing enterocolitis; NGS, next-generation sequencing; NLR, NOD-like receptor; PCA, principal component analysis; RNAseq, RNA sequencing; scRNAseq, single-cell RNA sequencing; SI, small intestine; SIP, spontaneous ileal perforation; smFISH, single-molecule fluorescence in situ hybridization; TCRβ, T-cell receptor beta; TLR, toll-like receptor.

all-cause mortality; however, mortality related to NEC has increased [3]. In addition to short-term complications, NEC is associated with high rates of long-term morbidity that include GI strictures, feeding intolerance, and short gut syndrome, but also significant systemic consequences such as microcephaly and neurodevelopmental delays [4,5].

NEC is a multifactorial disease involving environmental, microbial, host, and immune factors. However, despite intense research over the past several decades, the precise etiology of NEC continues to be elusive [6] and effective prevention methods or treatment options are unavailable. Dysregulation in both the mucosal immune system and the epithelial barrier are hypothesized to be associated with NEC, yet the mechanism of how these cells contribute to disease onset or progression is not clear. Previous studies have focused on specific cellular populations [7,8] and a comprehensive systems biology analysis is lacking.

With the goal of developing better treatments for premature infants with NEC, we reconstruct a single-cell atlas of neonatal and NEC small intestine (SI) tissue with an emphasis upon cellular localizations and interactions. Using single-cell RNA sequencing (scRNAseq), we define transcriptional signatures and ligand–receptor interactions associated with NEC. This is combined with deconvolution of bulk RNA sequencing (RNAseq) data for cell type abundance validation and imaging mass cytometry (IMC) to define cellular interactions. Our data demonstrate that aberrant cellular interactions are associated with NEC intestinal inflammation. Specifically, we describe an increase in inflammatory macrophages and inflammatory changes in conventional and regulatory T cells accompanied by increased T-cell receptor beta (TCRβ) clonality. There is profound remodeling of NEC mucosa characterized by a decrease in the proportion of top villus epithelial cells and an increase in proportion of endothelial cells and fibroblasts. All 3 cell types exhibit an increase in the transcription of inflammatory genes. Our NEC single-cell atlas identifies networks controlling intestinal homeostasis and inflammation thereby improving our understanding of NEC pathogenesis and identifying biomarkers and targets for therapeutics discovery.

## Results

### A single-cell atlas of human NEC and neonatal small intestinal samples

To characterize the cell states associated with NEC, we analyzed intestinal tissues from 19 patients with NEC and from 13 neonates that underwent intestinal surgery for non-NEC-related conditions. The samples were used in various assays listed in S1 Table and Fig 1A. The infants with NEC were significantly younger than the neonatal subjects (median gestational age at birth of 28 weeks in NEC versus 37 weeks in the comparison group, $p < 0.001$). However, the postnatal age of patients at the time of surgery was similar between the 2 groups (median age of 24.5 days in NEC versus 14 days in neonatal group, $p = 0.089$). Finding appropriate patients to serve as controls for comparison to samples from patients with NEC is challenging because healthy premature infants do not undergo intestinal surgery. Other controls beyond non-NEC-related neonatal tissue that have been historically used include fetal tissue, tissue from infants with spontaneous ileal perforation (SIP), and tissue obtained from re-connection surgery in infants recovered from NEC. Fetal tissue has not been exposed to an abundant microbiome and is likely immunologically different from that of postnatal samples. Infants with SIP, although gestationally age matched to those with NEC, usually undergo surgery in the first week of life and may have potential SIP-specific immune dysregulation [9]. Samples obtained from patients who have recovered from NEC (post-NEC surgeries) are obtained after prolonged periods of parenteral nutrition and antibiotic use both of which can drastically alter the mucosal tissue. As such, we opted to use non-NEC-related neonatal

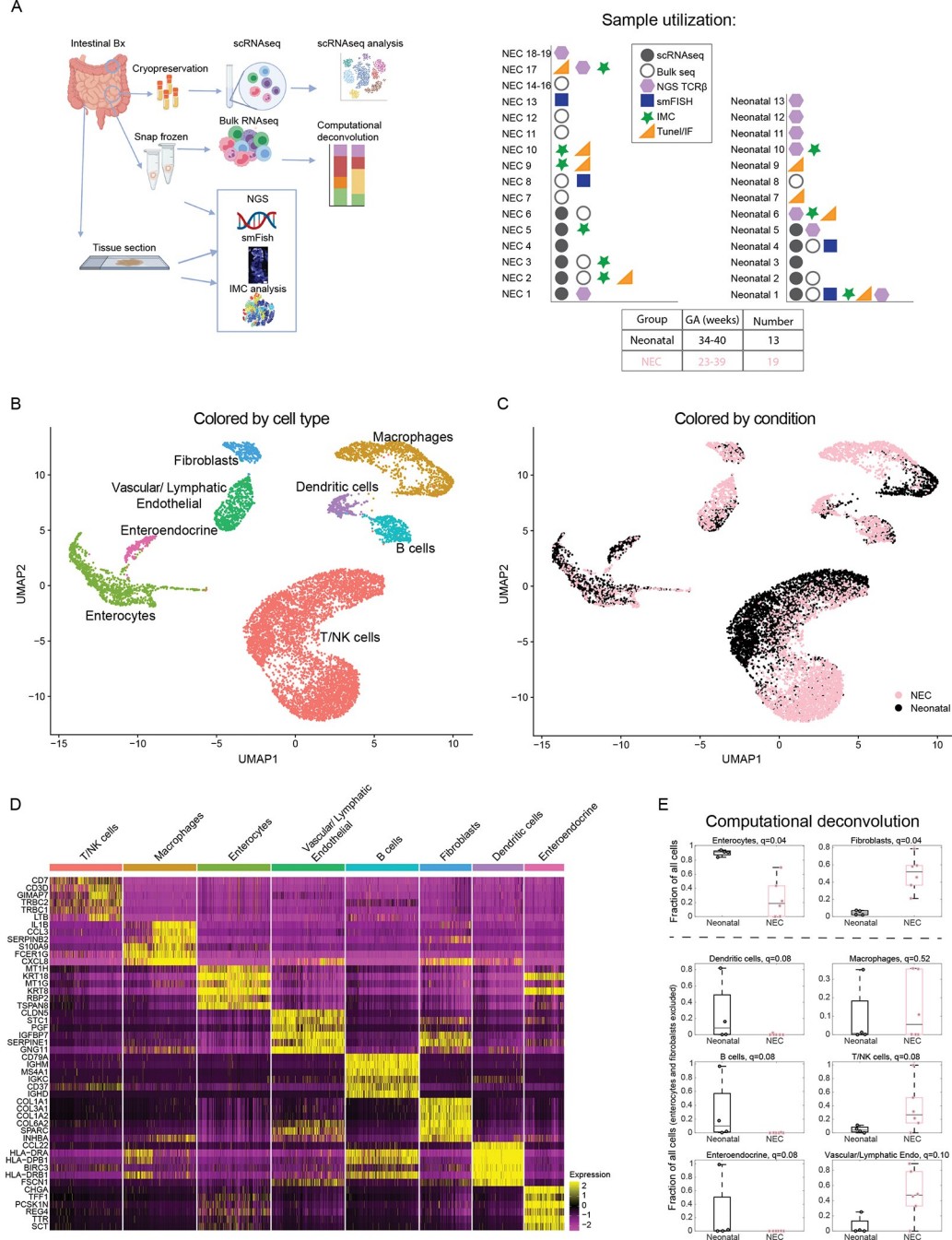

**Fig 1. A single-cell atlas of NEC and control human small intestinal samples.** (**A**) Experimental layout—human small intestinal tissues from neonates and NEC patients were harvested and used for scRNAseq, bulk RNAseq, smFISH, IMC, and NGS of the TCRβ. Samples used in various experiments listed on the right-hand side (**S1 Table**). (**B**) Single-cell atlas annotated by cell type. (**C**) Single-cell atlas annotated by condition. (**D**) Top 6 markers of the cell types in **B**. (**E**) Estimates of the proportion of enterocytes, fibroblasts, dendritic cells, macrophages, B cells, T-NK cells, enteroendocrine cells, and vascular/lymphatic endothelial cells based on computational deconvolution of the bulk RNAseq using the atlas single-cell populations (**Methods**). Each dot is a sample, fractions of enterocytes and fibroblasts normalized to the sum of cell fractions, remaining fractions normalized to the sum of all cells after excluding fibroblasts and enterocytes; q-values are computed based on FDR correction for all cell populations in the full atlas (**Methods**). Gray lines are medians, black/pink boxes are 25–75 percentiles. Only samples with Spearman correlations >0.3 between the mixture data and the synthetic mixtures are shown (neonatal: $n = 4$, NEC: $n = 6$). The data underlying this figure is available at the Zenodo repository under the following: https://doi.org/10.5281/zenodo.5813397 and in **S10 Table**. FDR, false discovery rate; IMC, imaging mass cytometry; NEC, necrotizing enterocolitis; NGS, next-generation sequencing; RNAseq, RNA sequencing; scRNAseq, single-cell RNA sequencing; smFISH, single-molecule fluorescence in situ hybridization; TCRβ, T-cell receptor beta.

samples as the comparison group as the best chronologically matched tissue without known immune disturbances.

We performed scRNAseq on 11 subjects (6 NEC and 5 neonatal samples, Fig 1B and 1C). This analysis enabled the extraction of gene expression signatures of the diverse epithelial, stromal, and immune cell populations. While single-cell atlases are powerful for extraction of expression signatures, estimation of population proportions from such datasets can be skewed due to differential viability of extracted cells [10]. To enable precise estimation of population proportions, we performed computational deconvolution of bulk RNAseq (6 NEC and 4 neonatal samples that passed threshold, Methods) data based on the clusters identified in the scRNAseq dataset (Fig 1A). We implemented this with next-generation sequencing (NGS) of TCRβ to identify clonality changes associated with NEC (4 NEC and 7 neonatal samples). Finally, we utilized IMC of 6 NEC and 3 neonatal samples to define niche cellular interactions enriched in NEC (Fig 1A). Overall, we generated a resource atlas that can enable the exploration of ligand–receptor interactions between any pairs of cell types.

Our atlas included 11,308 cells and revealed 8 main cell clusters representing macrophages, dendritic cells (DCs), B cells, T/Natural Killer (NK) cells, vascular/lymphatic endothelial cells, fibroblasts, enteroendocrine, and other enterocytes populations (Fig 1B–1D). Each cluster exhibited distinct gene expression markers (Fig 1D and S2 Table). We verified the stability of the expression signatures by reconstructing them based on subsampled patients and cells (S1A and S1B Fig). Our atlas enabled exploring the detailed gene expression changes in distinct cell subsets. Furthermore, bulk RNAseq analysis followed by computational deconvolution facilitated determination of major cluster abundances (Figs 1E and S1C) that revealed a substantial increase in the proportions of fibroblasts and a decrease in the proportions of enterocytes. Among the remaining cells, we identified a significant increase in the proportions of T/NK cells, vascular/lymphatic endothelial cells and a decrease in the proportions of DCs and B cells (Figs 1E and S1C, using false discovery rate (FDR) <0.25 as measure of significance).

## NEC is associated with an expression increase in inflammatory genes in macrophages and dendritic cells

Myeloid cells have been shown to be enriched in NEC tissue and to exhibit an increase in inflammatory programs [9,11–13]. Our atlas included 1,836 myeloid cells, which clustered into dendritic cells, noninflammatory macrophages, and 2 separate clusters of inflammatory macrophages (Fig 2A–2C). One of the inflammatory macrophage clusters, made up almost exclusively of NEC-associated macrophages (inflammatory macrophages group A), was marked by inflammatory cytokines and chemokines such as *IL6*, *IL1B*, and *CXCL8*, previously shown to be associated with NEC (Fig 2C) [6,9,14]. Accordingly, differential gene expression (DGE) between the NEC and neonatal macrophages showed a significant increase in proinflammatory molecules, including *IL1A*, *IL1B*, *IL6*, *CSF2*, *CSF3*, *CXCL8*, *CCL3*, *CCL4*, *CXCL2*, *CCL20* (associated with recruitment of leukocytes to the sites of inflammation) and signaling molecules such as *IRAK2*, *SOD2*, *NFKBIZ*, and *NFKBIA* (Fig 2C and 2D, Methods). NEC macrophages exhibited down-regulation of anti-inflammatory genes such as *CD9*, *MERTK*, *HLA-DRB5*, *TGFBI*, and *PLXDC2* [15–19]. Similarly, DCs in NEC up-regulated genes associated with inflammation such as *ISG15*, *S100A6*, *SERPINB1*, *GNG11*, *CCL17*, and *FSCN1* [20] and genes involved in DC activation—*CD40* and *CD44* [21,22] (Fig 2E and S3 Table). Gene set enrichment analysis (GSEA) revealed that macrophages were enriched in inflammatory gene sets, such as TNFα signaling via NF-κB, inflammatory response, NOD-like receptor (NLR) signaling, and toll-like receptor (TLR) signaling (Fig 2F). Both NLR and TLR activation have been implicated in NEC pathogenesis and NF-κB signaling in inflammation [23–25]. DCs in

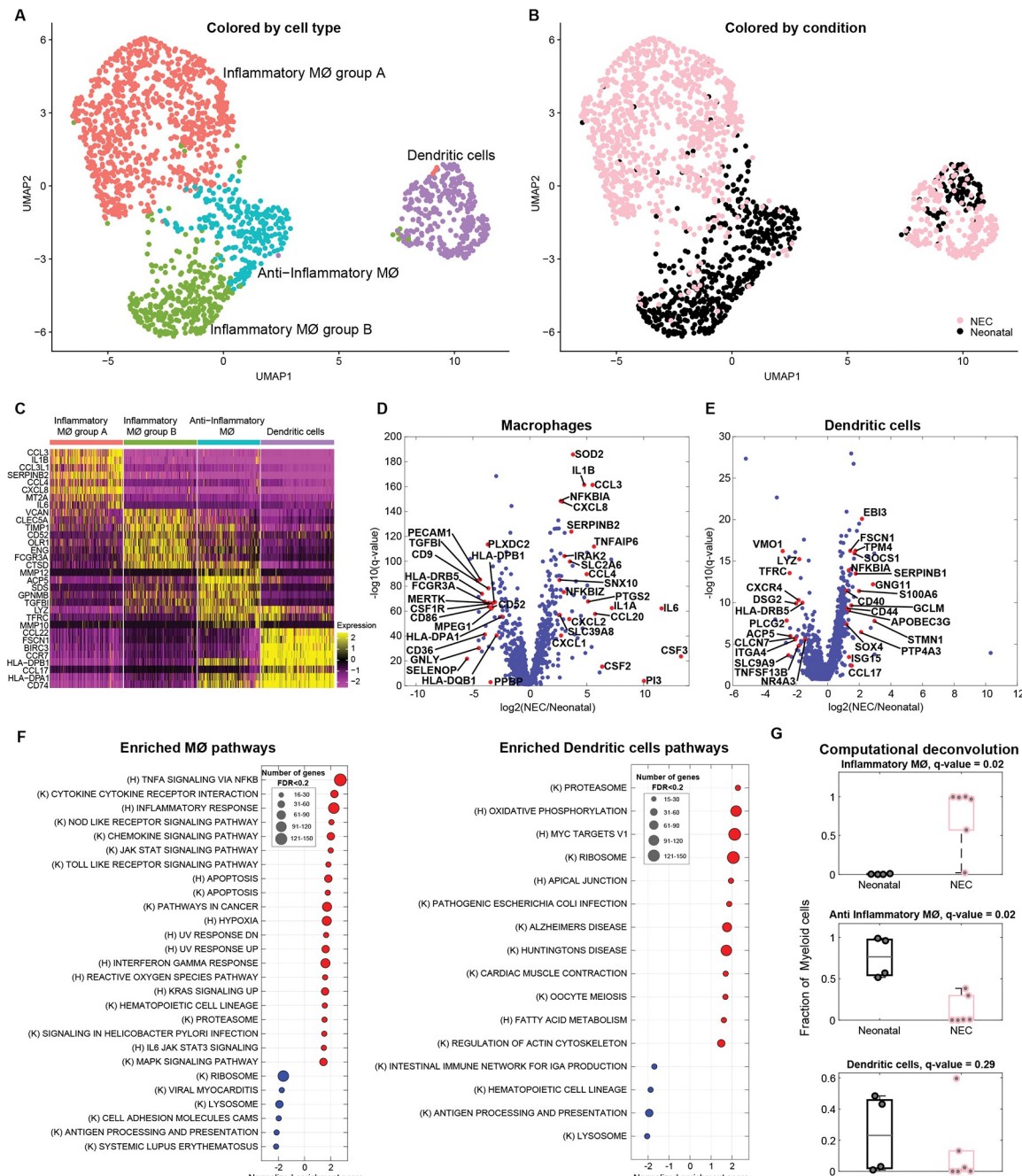

**Fig 2. Inflammatory macrophages are increased in NEC.** (**A**) Re-clustered atlas of myeloid lineages. Mϕ–macrophages. (**B**) Single-cell atlas annotated by condition. (**C**) Top 8 markers of the myeloid cell subtypes. (**D**) Differential gene expression between NEC and neonatal macrophages. (**E**) Differential gene expression between NEC and neonatal DCs. Red dots (D, E) are selected differentially expressed genes among the genes with q-value <0.02 and fold change above 3 or below 1/3. Included are all genes with sum-normalized expression above $10^{-4}$. (**F**) GSEA of pathways enriched (red) or depleted (blue) in NEC samples compared to neonatal samples for macrophages and DCs (q-value <0.2). (K) = Kegg pathways, (H) = Hallmark pathways. (**G**) Estimates of the proportions of distinct myeloid cell subsets based on computational deconvolution of bulk sequencing data. Each dot is a sample, proportions were renormalized over all myeloid cells, q-values are computed based on FDR correction for myeloid cells only (**Methods**). Gray lines are medians, black/pink boxes are 25–75 percentiles. Only samples with Spearman correlations >0.3 between the mixture data and the synthetic mixtures are shown (neonatal: *n* = 4, NEC: *n* = 6). The data underlying this figure is available at the Zenodo repository under the following: https://doi.org/10.5281/zenodo.5813397 and in **S3 and S10 Tables**. DC, dendritic cell; FDR, false discovery rate; GSEA, gene set enrichment analysis; NEC, necrotizing enterocolitis.

NEC were enriched in MYC target genes and ribosomal pathways (Fig 2F). Myc signaling in DCs has been shown to be critical for optimal T cell priming [26]. Computational deconvolution of the bulk RNAseq samples revealed a nonsignificant trend towards an increase in the proportion of all macrophages (fold change = 7.7, q = 0.52, Figs 1E and S1C), an increase in the inflammatory macrophages (fold change = 667, q = 0.02, Fig 2G) and a decrease in the noninflammatory macrophages (fold change = 0.009, q = 0.02, Fig 2G) and a nonsignificant decrease in the proportion of DCs in the NEC samples (fold change = 0.056, q = 0.29, Fig 2G, using FDR < 0.25 as measure of significance). Notably, although the fraction of DCs in NEC was lower than in neonatal samples, their expression signatures showed elevation of proinflammatory genes, similarly to the NEC macrophages.

## T-cell proportions, clonality, and characteristics are changed in NEC

The contribution of lymphocytes to the pathogenesis or progression of NEC has been controversial. Some studies have reported an increase in the abundance of T cells while others have shown a decrease [27–29]. Moreover, work from the Hackam group has suggested that Th17 lymphocytes are critical to NEC pathogenesis [30,31]. Our atlas included 6,400 T/NK cells, with 7 distinct T and NK cell subsets including: NK cells, innate lymphoid cells (ILCs), naïve T cells, regulatory T cells (Tregs), T helper cell 1 (Th1), proliferating T cells, and activated T cells (Figs 3A and 3B and S2A).

The T-cell receptor repertoire is crucial in adaptive immunity and unique sequences constitute diversity. To interrogate the TCR repertoire and determine clone-size distribution and gene usage, we performed NGS of the TCRβ in NEC and neonatal SI and large intestinal (LI) samples. Overall, TCRβ clonality was increased in NEC compared to neonatal samples (Fig 3C). Additionally, analysis of gene usage revealed that the use of variable (V), diversity (D), and joining (J) segments differed between NEC and neonatal cases (Fig 3D) with an increased frequency of TRBV10 and decreased use of TRBV15, TRBJ1-4, and TRBJ2-1 in NEC compared to neonatal cases (S2B Fig). NEC patients had shorter CDR3β length with fewer deletions and fewer insertions than neonatal controls (S2C Fig), a phenomenon previously reported in IBD [32].

To identify if the clones expanded in NEC bind to known epitopes, we performed a search of the public clones database. Public clones have a unique amino acid or nucleotide sequence, are present across individuals, and can be readily identified in a published database (https://vdjdb.cdr3.net/). The majority of the top clones observed in NEC were previously reported public clones. Our search revealed only 3 public clones that were significantly enriched in NEC. Interestingly, the amino acid sequences of 2 of these clones had an identical sequence to clones known to bind to cytomegalovirus (CMV) (CMV-1, CMV-2) (S2D Fig and S4 Table).

Computational deconvolution analysis of the bulk RNAseq data indicated a nonsignificant increase in activated T cells, and a significant increase in Th1 cells and proliferating T cells and a decrease in naïve T cells (S2E Fig, using FDR <0.25 as measure of significance). To investigate if T cells, ILCs, and NK cells were transcriptionally altered in NEC, we compared the transcriptomes of all T-cell clusters between NEC and neonatal samples. DGE showed an increase in an inflammatory signature in all subtypes except the naïve T cells that were transcriptionally similar between the groups (Fig 3E). The ILC cluster had an increase in *CCR7*, a gene associated with trafficking (Fig 3E), while the NK cluster had an increase in *CCL3* [33], *CD83*, *XCL*, *GZM*, and TNF-related genes (Fig 3E). Similarly, the activated T-cell cluster showed an up-regulation of inflammatory and cytotoxic genes such as *GZMA/B*, *GNLY*, *KLRC1*, *CSF2*, *IL23A*, *CXCL8*; genes indicative of activation such as *IL2RA*; and genes downstream of IFN-γ signaling such as *IRF1*, *GBP5*, and *STAT* [34] (Fig 3E). We could not detect *IL17A* expression in our

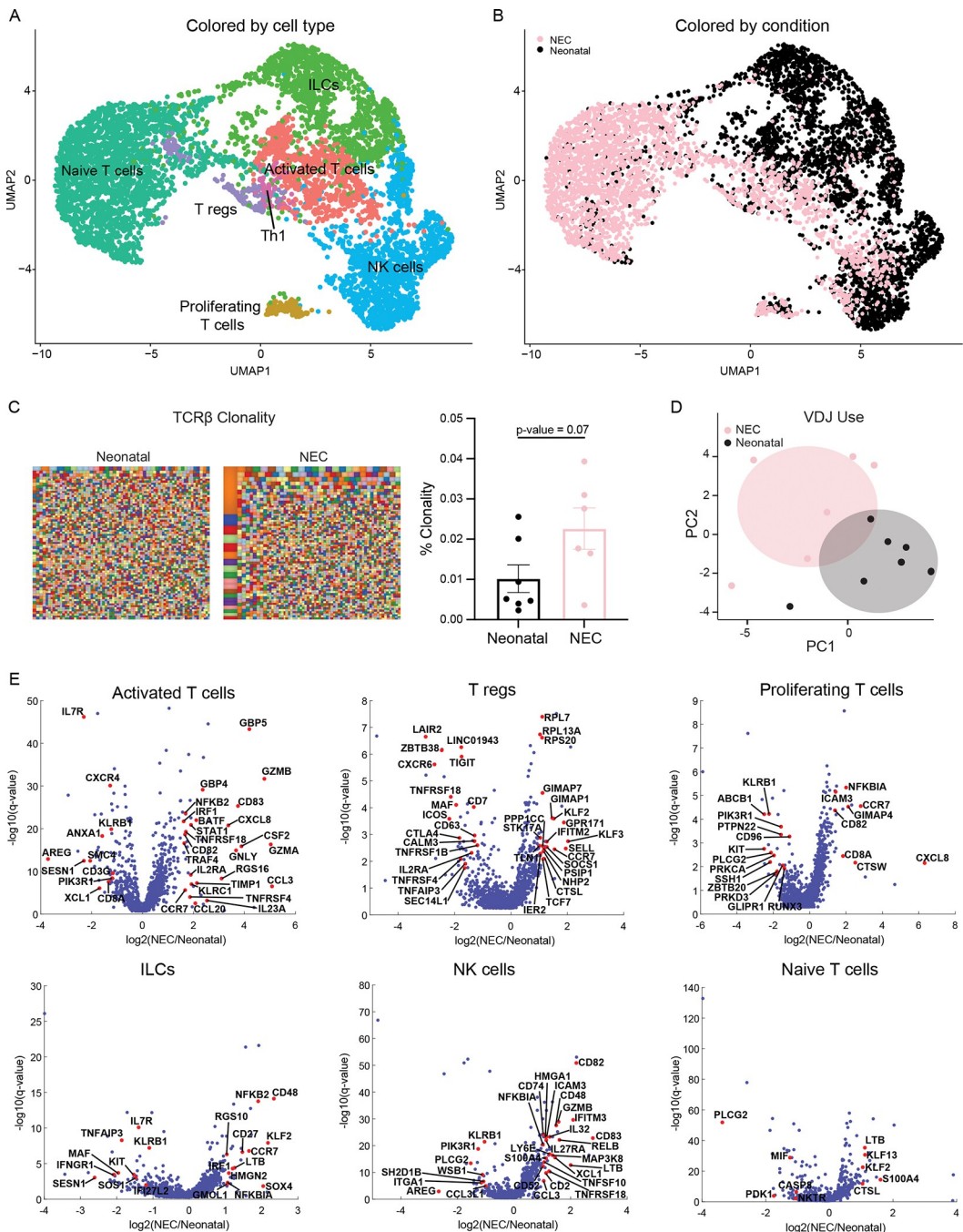

**Fig 3. Landscape and transcriptional signatures of T/NK/ILCs populations in NEC.** (**A**) Re-clustered atlas of the T/NK cluster. (**B**) Single-cell atlas annotated by condition. (**C**) NGS of TCRβ candy plots where each small square represents 1 clone with the squares proportional to the number of T cells with a particular clone with quantification on the right. Each dot represents 1 tissue sample (neonatal: $n = 7$, NEC: $n = 6$, from 7 neonatal and 4 NEC patients). (**D**) PCA plot of variable (V), differential (D), and joining (J) regions use in NEC and neonatal cases (neonatal: $n = 7$, NEC: $n = 6$, from 7 neonatal and 4 NEC patients). Shaded areas are 95% confidence intervals. (**E**) Differential gene expression between NEC and neonatal T-cell populations. Red dots are selected differentially expressed genes among the genes with q-value <0.02 and fold change above 2 or below 1/2. Included are all genes with normalized expression above $10^{-4}$. The data underlying this figure is available at the Zenodo repository under the following: https://doi.org/10.5281/zenodo.5813397 and in **S3 Table**. ILC, innate lymphoid cell; NEC, necrotizing enterocolitis; NGS, next-generation sequencing; PCA, principal component analysis; TCRβ, T-cell receptor beta.

atlas and *IL17F* and *IL22*, cytokines also produced by Th17 cells, were only up-regulated in 1 NEC case and did not meet the threshold to be included in the DGE analysis. However, a number of other IL17 signature genes were up-regulated in activated T cells associated with NEC including *CCL20*, *TIMP1*, and *BATF* [35] (Fig 3E). We could not perform DGE for the Th1 cluster as there were very few contributing neonatal T cells. In summary, T cells in NEC show increased clonality and expression of proinflammatory genes compared to the non-NEC mucosa.

## Regulatory T cells with reduced immunosuppressive signature are associated with NEC

Tregs have an important role in regulating mucosal homeostasis. Various reports have hypothesized different roles of Tregs in the pathogenesis of NEC [29,36,37]. Our computational deconvolution showed no difference in the proportion of Tregs between NEC and neonatal tissue (S2E Fig). However, DGE analysis of the Tregs revealed an increase in *SELL* (CD62L), *CCR7*, as well as *SOCS1*. *CCR7* expression suggests that there are more recent thymic emigrants or natural Tregs in NEC mucosa. Furthermore, *SOCS1* is essential for maintaining T reg suppressive function [38](Fig 3E and S3 Table). Interestingly, there was a reduction in the expression genes classically associated with Treg identity or suppression function such as *CTLA-4*, *TIGIT*, *iCOS*, *TNFRSF4* (encoding the Ox40 receptor found on intestinal Tregs and effector memory T cells), and *IL2RA* (Fig 3E and S3 Table). Our analysis indicates no changes in the proportion of Tregs, but they exhibit a phenotype of reduced suppressive activity and could represent impaired T-reg function in NEC.

## NEC vascular and lymphatic endothelial cells exhibit distinct proinflammatory signatures

Our single-cell atlas included 407 vascular endothelial and 332 lymphatic endothelial cells, each clustering into proinflammatory and noninflammatory subsets (Figs 4A–4C and S3A and S3B). Upon DGE analysis, we found that NEC vascular and lymphatic endothelial cells uniformly up-regulated chemokines such as *CCL2*, *CXCL1*, *CXCL2*, and *CXCL3* as well as adhesion molecule, *ICAM1*, consistent with endothelial activation and leukocyte recruitment (Fig 4D and 4E) [39]. In vascular endothelial cells, we observed up-regulation of adhesion molecules *SELE* [40], procoagulant factors such as *SERPINE1* and *F3* along with changes in regulators of blood flow that reduce perfusion including increased *EDN1* and decreased *NOS3* (Fig 4D). We observed significant up-regulation of cytokines and chemokines in lymphatic endothelial cells including *IL6*, *CCL5*, *CXCL10* [41], *CSF2*, *CSF3*, *CCL20*, *CXCL8* and proinflammatory signaling genes such as *TNFAIP2*, *TNFAIP6*, *TNFAIP8*, *TRAF1* (where binding to TNFR2 enhances CXCL8 production [42]), and *RIPK2* (Fig 4E). Pathway analysis revealed both vascular and lymphatic endothelial cells were significantly enriched in NFB-responsive genes, chemokine signaling pathway, and IFNγ responses (Fig 4F and 4G). Both *IL1B* and *TNFA* were enriched in the NEC inflammatory macrophages and were associated with an increase in *SELE* and *ICAM1* in the activated endothelium in NEC (S3C and S3D Fig). Apoptotic gene sets were also enriched in both vascular and lymphatic endothelial cells (Fig 4F and 4G) and were confirmed by an increase in TUNEL staining in LYVE-1$^+$ endothelial cells (Fig 4H and 4I). Our computational deconvolution suggested an overall increase in the proportions of vascular/lymphatic endothelial cells (Figs 1E and S1C). In line with the elevation in apoptotic markers, the elevated cell proportions could be a result of compensatory elevated proliferation, balancing the elevated cell death. Indeed, as 4.4% versus 0.7% of vascular/lymphatic cells in NEC/neonatal, respectively, were positive for the mRNA of the proliferation marker Mki67

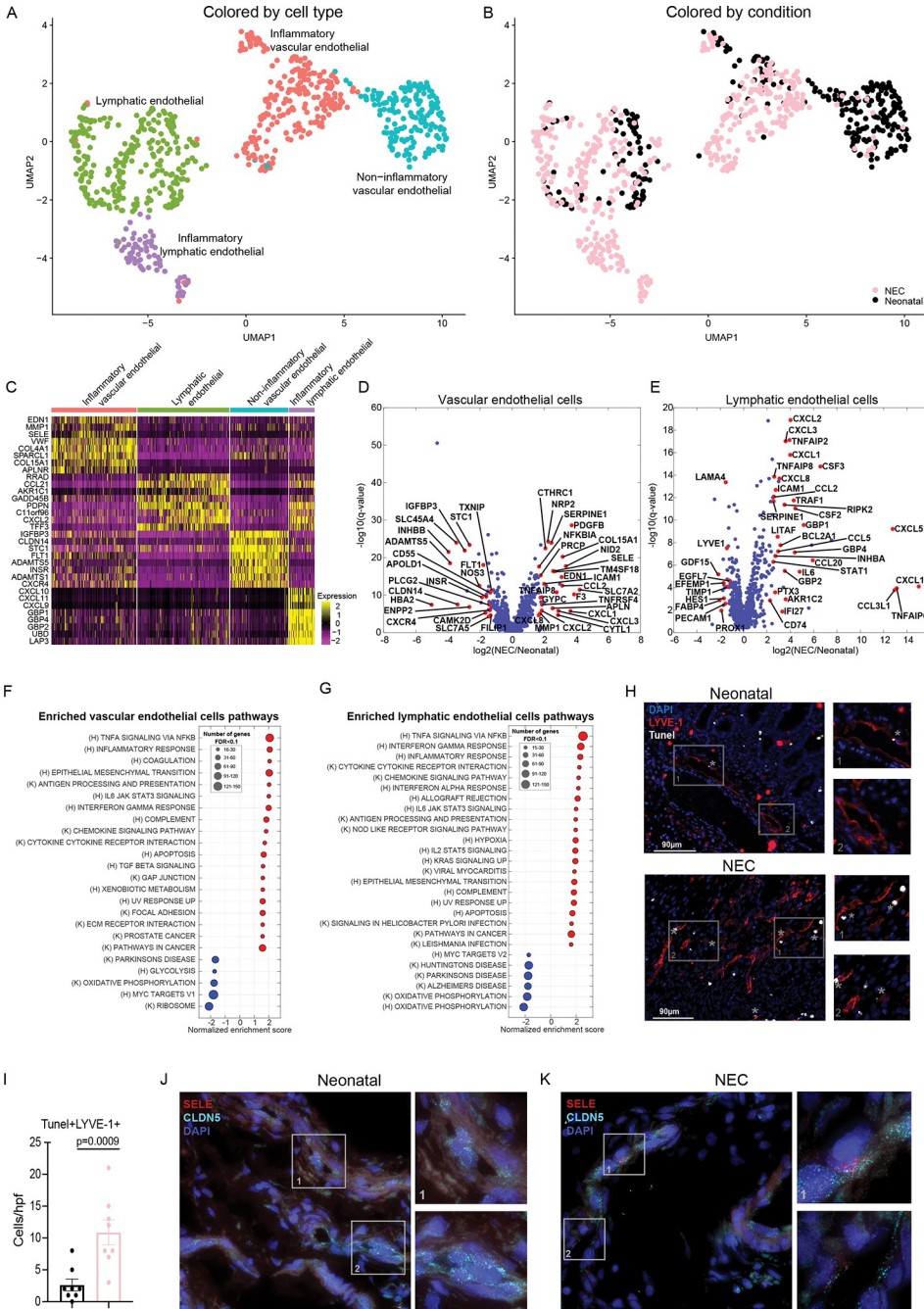

**Fig 4. Lymphatic and vascular endothelial cells in NEC exhibit proinflammatory signatures.** (**A**) Re-clustered atlas of the lymphatic and vascular endothelial cluster. (**B**) Single-cell atlas annotated by condition. (**C**) Top 8 markers for the cell types in **A**. (**D, E**) Differential gene expression between NEC and neonatal cells for vascular endothelial cells (D) and lymphatic endothelial cells (E). Included are all genes with sum-normalized expression above $10^{-4}$. Red genes are selected differentially expressed genes among the genes with q-value <0.02 and fold change above 2 or below 1/2. (**F, G**) GSEA of pathways enriched (red) or depleted (blue) in NEC samples compared to neonatal samples for vascular endothelial (F) and lymphatic endothelial cells (G) with q-value <0.1. (K) = Kegg pathways, (H) = Hallmark pathways. (**H**) Representative immunofluorescence images from neonatal and NEC samples stained with LYVE-1 (red) and TUNEL staining (white). Gray asterisk (*) represents apoptotic endothelial cells (Tunel+LYVE-1+). Scale bar: 90 μm. (**I**) Quantification of Tunel+LYVE-1+ cells. Each dot represents 1 image, 2 images/sample (neonatal: $n = 4$, NEC: $n = 4$). (**J, K**) smFISH demonstrating increase in SELE+ endothelial cells in NEC. Red dots are individual mRNAs of SELE; cyan dots are individual mRNAs of CLDN5, a marker of vascular/lymphatic endothelial cells. Blue are DAPI-stained nuclei; scale bar: 10 μm. J and K are representative images from $n = 2$ subjects per group. The data underlying

this figure is available at the Zenodo repository under the following: https://doi.org/10.5281/zenodo.5813397 and in **S10 Table**. GSEA, gene set enrichment analysis; NEC, necrotizing enterocolitis; smFISH, single-molecule fluorescence in situ hybridization.

(Fisher exact test $p$ = 0.0032). To validate the induction of proinflammatory endothelial cells in NEC, we performed single-molecule fluorescence in situ hybridization (smFISH) for the endo-thelial/lymphatic-specific marker *CLDN5* along with *SELE* as a specific marker of endothelial activation, and identified elevation in SELE+ endothelial cells in NEC (Fig 4J and 4K). In sum-mary, we found a proportional increase in endothelial cells in NEC with a shift in vascular and lymphatic endothelial cells towards a proinflammatory state, complemented by increased coagulation and decreased perfusion-associated genes in the vascular endothelium.

## NEC epithelial cells and fibroblasts exhibit an increased inflammatory potential with a loss of villus-tip enterocytes

We next turned to analyze the changes in enterocyte cell identity in NEC. Our atlas included 1,203 enterocytes (Fig 5A and 5B), which we further classified by crypt-villus zones using land-mark genes, such as the villus-top gene APOA4 and the crypt gene LGR5 (Fig 5C–5E). Computational deconvolution exposed an overall reduction in enterocytes (fold change = 0.2, q = 0.04, Figs 1E and S1C) with a reduction in villus top cells (fold change = 0.3, q = 0.2, Fig 5F) and an increase in crypt cells in NEC compared to the non-NEC comparison group (fold change = 76, q = 0.2, Fig 5F). IMC data also demonstrated villus blunting (Fig 5G). DGE of the lower villus zone revealed that NEC epithelial cells exhibited substantial increase in the antimi-crobial genes *LCN2*, *REG1A*, *REG1B*, and *DMBT1* (Fig 5H), genes previously shown to be ele-vated in inflamed epithelium associated with inflammatory bowel disease (IBD) [43,44]. NEC enterocytes further increased the expression of the chemokines such as *CXCL1*, *CXCL3* [45], *CXCL5*, and *CXCL8* (Fig 5H and S3 Table). Likewise, expression of *DUOXA2*, an IBD-associ-ated gene that produces $H_2O_2$ [46] was increased in NEC epithelial cells. There was increased expression of STAT3 target genes: *REG1A*, *REG1B*, *LCN2*, *DMBT1*, *CXCL5*, and STAT1 target gene *DUOXA2* [47] (Fig 5H). Notably, signature of TLR4 activation was enriched in NEC compared to neonatal epithelium (Fig 5I), consistent with previous reports of overactivation of the pathway in samples with NEC [23,30,48]. This increase in TLR4 pathway activation (Meth-ods) is likely due to an increase in the proportions of bottom and crypt villus zones, and indeed, TLR4 pathway activation was not differentially expressed when stratifying for crypt/mid-bottom enterocytes. Our atlas did not contain sufficient amounts of goblet cells, prohibit-ing analysis of potentially differentially expressed genes in these cells.

Clustering of the fibroblasts demonstrated a small neuronal population and 319 fibroblasts (S4A and S4B Fig). The low cell number of fibroblasts was likely an artifact of the processing of the tissue for the scRNAseq analysis. Using computational deconvolution, we identified an increase in the overall fibroblasts in NEC tissue (fold change = 11.7, q = 0.04, Figs 1E and S1C). NEC fibroblasts exhibited an increase in several inflammatory genes including *IL1B*, *CSF2*, *CSF3*, *EREG*, and *CCL20* (S4C Fig and S3 Table).

## Epithelial–mesenchymal–immune interactions are altered in NEC

To understand if cellular interactions are altered in NEC in the native state, we applied IMC [49]. In this method, formalin-fixed paraffin-embedded sections of small intestine (6 NEC and 3 neonatal samples) are incubated with a cocktail of heavy metal chelated antibodies (Methods, S5 Table), ablated and analyzed. Clustering analysis of 20,819 cells from IMC data revealed

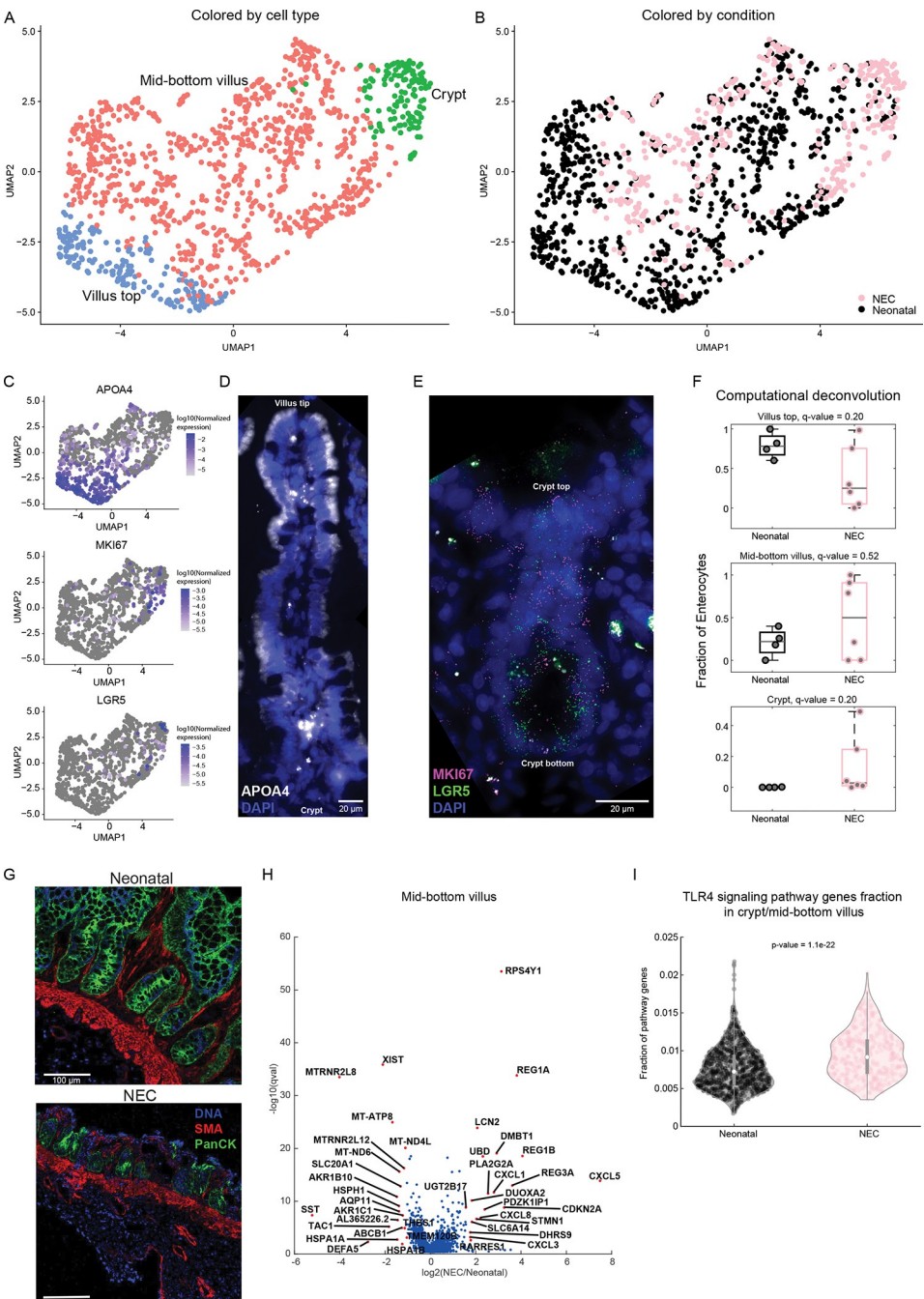

**Fig 5. Changes in enterocyte gene expression and zonal representation in NEC.** (**A**) Re-clustered atlas of the enterocyte cluster colored by crypt-villus zone. (**B**) Single-cell atlas annotated by condition. (**C**) UMAPs colored by top villus marker–APOA4, proliferation marker–MKI67, and stem cell marker–LGR5. Color bar is log10 (normalized expression). (**D**) smFISH of epithelial cells demonstrates increase in APOA4+ (white) epithelial cells towards the top of the villus. (**E**) smFISH of crypt cells, magenta dots are individual mRNAs of MKI67, green dots are individual mRNAs of LGR5 in the crypts. Blue are DAPI-stained nuclei, scale bars: 20 μm. (**F**) Estimates of the proportions of villus-crypt zones subsets based on computational deconvolution of bulk sequencing data. Each dot is a sample, proportions were renormalized over all villus-crypt zones, q-values are computed based on FDR correction for enterocytes only (**Methods**). Only samples with Spearman correlations >0.3 between the mixture data and the synthetic mixtures are shown (neonatal: *n* = 4, NEC: *n* = 6). (**G**) Representative images from Histocat 1.7.6.1 showing villus blunting in NEC compared to neonatal tissue. DNA– 191/193-intercolator (blue), SMA- smooth muscle actin (red), panCK-pancytokeratin (green). (**H**) Differential gene expression between NEC and neonatal cells for the mid-bottom villus zone. Included are all genes with sum-normalized expression above 5 × 10⁻⁵. Red dots are the top 20 most

differentially expressed genes among the genes with q-value <0.02 and fold change above 2 or below 1/2. (**I**) TLR4 gene signature in NEC and neonatal samples. *P*-value calculated using two-sided Wilcoxon rank-sum test. The data underlying this figure is available at the Zenodo repository under the following: https://doi.org/10.5281/zenodo.5813397 and in **S3 and S10** **Tables**. FDR, false discovery rate; NEC, necrotizing enterocolitis; smFISH, single-molecule fluorescence in situ hybridization.

numerous clusters including epithelial cells, vascular endothelial cells, fibroblasts, B cells, T cells, macrophages, DCs, and clusters of cells that could not be identified with the markers used (other, S5A and S5B Fig).

To define the cellular interactions in NEC small intestine, we performed a nearest neighbor analysis of IMC data using histoCAT, a tool that identifies statistically significant interactions/avoidances between cellular clusters (Methods) [50]. Numerous cell type interactions were altered in NEC (Fig 6A and S6 Table). Consistent with up-regulation of genes associated with leukocyte recruitment on vascular endothelial cells such as SELE (Fig 4D and 4K), NEC tissue had increased interactions of vascular endothelial cells with DCs. Additionally, fibroblasts had increased interactions with several cellular populations including epithelial cells and monocytes/macrophages and B cells. Consistent with increase in T-cell clonality, NEC small intestine showed significant interactions between memory T cells and antigen presenting cells including DCs and macrophages. Finally, Tregs had significantly altered interactions with several other cell types including decreased interactions with CD16+ macrophages (Fig 6A).

To identify the chemokines and cytokines that could be associated with recruitment of inflammatory macrophages and lymphocytes or could be responsible for altered signaling within the NEC small intestine, we performed a ligand–receptor analysis between all pairs of cell types in our atlas (S7 Table, Methods). To this end, we parsed a database of ligands and matching receptors [51] and defined an interaction potential between each pair of cell types as the product of the ligand expression in the sender cell type and the expression of matching receptor in receiving cell type. This potential was computed separately for the NEC cells and the neonatal cells, and the ratios of interaction potentials between NEC and neonatal samples were statistically assessed via random re-assignment of cells to the 2 groups (Methods). Consistent with increased activation of endothelial cells in our scRNAseq data, we found that NEC vascular endothelial cells up-regulated signaling to leukocytes via integrin receptors and cytokines including: macrophages via VCAM1-ITGB1 and IL6-F3 (Fig 6B), dendritic cells via VCAM1-ITGB1/ITGB7, and IL6-IL6ST (Fig 6C), and naïve T cells via MADCAM1-ITGB7, VCAM1-ITGB1/ITGB7/ITGB2/ITGA4, and IL6-IL6R/IL6ST (Fig 6D). Similarly, interactions of vascular endothelial cells through VCAM1-ITGB1/ITGB7/ITGA4/ITGB2 were up-regulated in all T-cell subsets including activated and regulatory T cells (S5C Fig) and MADCAM1-ITGB7 were additionally up-regulated in Tregs (S5C Fig). NEC lymphatic endothelial cells, primarily interacted with leukocytes via cytokines and chemokines including: to macrophages via CCL3L1-CCR1 (Fig 6E), dendritic cells via CXCL9/CXCL10/CXCL11-CXCR3 and CCL20-CCR6 (Fig 6F), and naïve T cells via CXCL9/CXCL10/CXCL11-CXCR3 (Fig 6G). Our ligand–receptor analysis indicated that NEC epithelial cells interact with macrophages through elevated levels of TNFSF9 (Fig 6H), and NK cells signal to NEC epithelial cells via IL1B-IL1R1/IL1R2 and GZMB-PGRMC1 (Fig 6I). Fibroblasts interacted with macrophages through CXCL10-SDC4 and IL15-IL2RA/IL15RA (Fig 6J). Finally, regulatory T cells interacted with both macrophages and fibroblasts through IL1B-IL1R1 and TNFSF14-TNFRSF14 (S5D Fig). Numerous other interactions between cell types were identified (S5E–S5G Fig), including interaction between Treg IL1B and epithelial IL1R1 and IL1R2 and activated T cells and epithelial cells via GZMB-CHRM3 and ICAM1-EGFR (S5E Fig). Overall, epithelial–mesenchymal–endothelial–immune interaction were altered in NEC with increased ligand–receptor

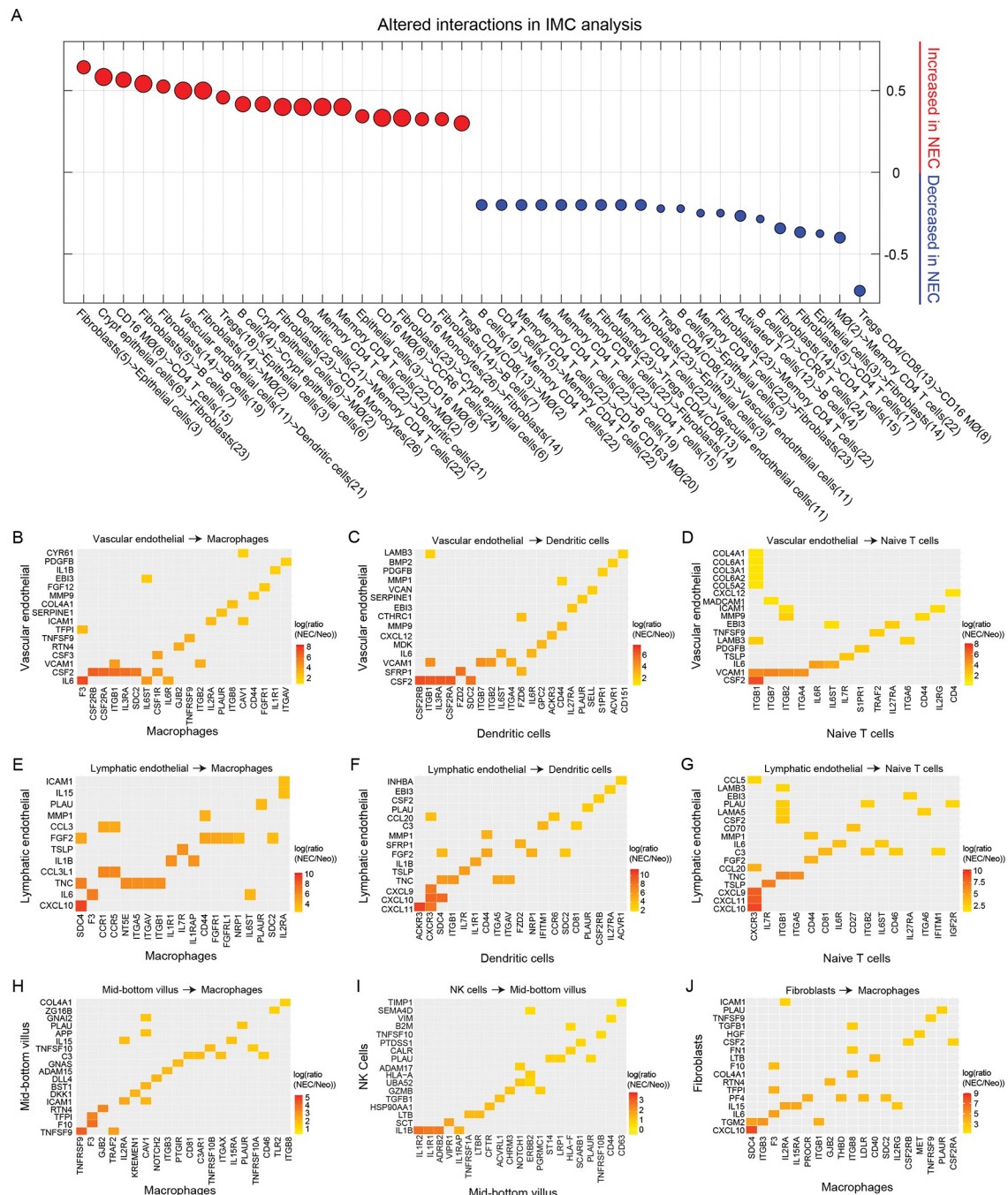

**Fig 6. Altered cellular adjacencies and protein–ligand interactions in NEC.** (**A**) Dot plot showing the 20 interaction types that have the highest increase (red) or decrease (blue) interaction values between neonatal (*n* = 3) and NEC (*n* = 6) samples (**Methods**). Interactions values (**S6 Table**) were computed by IMC analysis using Histocat 1.7.6.1 and 999 permutations and a *p*-value <0.01 [50]. Dot size corresponds to the interaction values in NEC. (**B–D**) Significantly elevated molecular interactions between vascular endothelial cells and macrophages (B), dendritic cells (C), naïve T cells (D). (**E–G**) Significantly elevated interactions between lymphatic endothelial cells and macrophages (E), dendritic cells (F), and naïve T cells (G). (**H**) Interactions between mid-bottom villus cells and macrophages. (**I**) Interactions between NK cells and mid-bottom villus cells. (**J**) Interactions between fibroblasts and macrophages. Shown are 16–25 significant interactions (q-value <0.01) with highest fold change (**Methods**). In all interaction, maps sender population is on the y-axis, receiver population is on the x-axis. (B–J) Neo = Neonatal. The data underlying this figure is available at the Zenodo repository under the following: https://doi.org/10.5281/zenodo.5813397 and in **S6 and S7 Tables**. IMC, imaging mass cytometry; NEC, necrotizing enterocolitis.

interactions between endothelial cells and immune cells, fibroblasts and immune cells, and epithelial cells and macrophages, T cells, and regulatory T cells.

## Discussion

The pathogenesis of NEC remains poorly understood and to date, few studies utilize single-cell approaches to describe the altered interactions among various intestinal cells [9]. Here, we systematically explored the changes in cell proportions and expression signatures in NEC using single-cell approaches. We present a single-cell atlas with spatial resolution of the small intestine from neonates with and without NEC that identified 8 distinct cellular populations: macrophages, DCs, B cells, T/NK cells, vascular/lymphatic endothelial cells, fibroblasts, enteroendocrine cells, and enterocytes. Our data exposes substantial changes in both abundances and transcriptional profiles of immune and non-immune populations in NEC small intestine.

NEC mucosa was marked with inflammatory changes in numerous innate immune populations. We observed an increased proportion of inflammatory macrophages that expressed inflammatory cytokines associated with lymphocyte recruitment including IL1A and IL1B and TNFA [52]. Furthermore, pathway analysis was suggestive of inflammatory responses including TNFa signaling via NFkB, cytokine receptor interactions, NLR and TLR signaling. NLR and TLR activation have been implicated in NEC pathogenesis both in human disease and in murine models, with the current study providing a context that macrophages are responding to these signals [23–25] and may be interacting with both immune and non-immune cells within the small intestine.

From the adaptive immune side, consistent with previous work that implicated Th17 T cells in the pathogenesis of NEC [30,31,53], we observed an up-regulation of genes classically expressed by Th17 cells (*BATF* and *CCL20* [54,55]) and additionally identified increased cytotoxic activity in NEC-activated T cells. Tregs are critical for maintaining homeostasis and play a major role in intestinal inflammation through cell–cell interactions and secreted factors. Previous studies showed conflicting results regarding the change in proportions of Tregs in NEC [36,37,53,56]. Similar to a recent study demonstrating heterogeneity between NEC samples [11], we observed large variability in the proportion of Tregs in NEC tissue. Our findings were indicative of a decrease in Treg suppressive abilities including down-regulation of genes associated with Treg identity or function (CTLA4, ICOS, TIGIT, and TNFRSF4, Fig 3E), overall alterations in the interactions of Tregs with other cell types including a reduction in the interactions with CD16+ macrophages (Fig 6A and S6 Table), and up-regulation of IL1B-IL1R interactions (S5D and S5E Fig) previously shown to render Tregs less suppressive [57]. As such, altered Treg function likely contributes to the progression of inflammation in NEC.

Surprisingly, major differences in TCRβ clonality, VDJ use, CDR3β length, and presence of shared clones were evident in NEC compared to controls. Specifically, we noted an increase in the frequency of public clones known to bind CMV antigens. It is unlikely related to the presence of active intestinal CMV infection, as these clones were present in samples from all patients with NEC and in majority of healthy fetuses [58]. Public clones are known to be promiscuous [59] and these could represent clones against bacterial or self-antigens that are similar to peptides found on CMV. We also noted shorter CDR3β length with fewer deletions and fewer insertions in NEC compared to the neonatal non-NEC comparison group, similar to what has been reported in IBD [32]. In the future, analysis of paired alpha/beta TCR chain analysis could further define specific clone/antigen interaction to characterize the role of clonal expansion in NEC pathogenesis.

Our investigation also revealed inflammatory signaling in non-immune cell types. Villus-top enterocytes were significantly reduced in NEC. A similar phenotype was observed in a

"gut on a chip" model of NEC [60,61], as well as in a recent study of samples from patients with NEC [62]. Enterocytes that were present in NEC exhibited an inflammatory phenotype, with up-regulation of chemokines involved in the recruitment of immune cells, and enrichment of ligand–receptor interactions between epithelial cells and immune cells. Furthermore, this was associated with an up-regulation of STAT3-dependent genes including *REG1A* [63], *REG1B*, *LCN2* [64], and *PLA2G2A*, antimicrobial peptides that are up-regulated upon epithelial damage [65] and shown to be up-regulated in IBD [43–46]. Similarly, fibroblasts demonstrated an increase in inflammatory genes (*IL1B* and *CSF3)* and extensive interactions with immune cells both by ligand–receptor interactions and by IMC neighborhood analysis. Previous work had identified inflammatory fibroblasts as potential drivers of IBD [66]. It is intriguing to postulate that a similar mechanism might be at play in NEC.

The proportion of inflammatory endothelial cells was also elevated in NEC-affected small intestine. Vascular endothelial cells up-regulated genes involved in vasoconstriction (*EDN1*), clotting (*F3*, *SERPINE1*) angiogenesis (*CXCL1*, *CXCL8*, and *ALPN*), proliferation [67], leukocyte recruitment (*SELE*), leukocyte adhesion (*ICAM1*, Fig 4D), and inflammatory pathways (TNFa and IFNγ signaling, Fig 4F). Our interaction analyses further revealed elevated ligands–receptor pairs involving the vascular adhesion molecules VCAM1 and MADCAM1 (Fig 6B–6D and S7 Table). Integrins interacting with VCAM1 and MADCAM1 include the α4β7 (ITGA4/ITGB7) complex that is up-regulated on activated lymphocytes, and innate cells, leading to leukocyte extravasation into intestinal high endothelial venules [68–71]. Up-regulation of α4β7 and α4β1 is thought to be pathogenic in IBD [72] and these ligand receptor interactions were enriched for in NEC samples. Blockade of α4β7 with Vedolizumab, a humanized monoclonal antibody against α4β7, is effective for induction and maintenance of remission in IBD in multiple studies [73–77]. It remains to be seen whether the long latency of action of this drug would still render it effective to the prevention and treatment of NEC. Overwhelming inflammation in endothelial cells may lead to dysfunction or death and contribute to intestinal inflammation. Indeed, pathway analysis of DGE in NEC endothelial cells showed apoptosis as one of the up-regulated pathways, a finding which we confirmed by TUNEL staining.

NEC is a progressive disease and our atlas captured dysregulation in the small intestine in the subset of infants that required surgery. One limitation of our study is that it did not include infants who recovered from NEC with medical therapy. Validation that some of the markers identified here can also be detected in the blood of infants with NEC and their identification in infants with various stages of NEC would be interesting. Additionally, understanding the role of the dysbiosis in NEC pathogenesis by including biopsy-associated microbiome data could be the focus of future studies.

In summary, we provide a comprehensive atlas of cellular dysregulation in NEC, accompanied by localization and ligand–receptor interaction analysis. Our study demonstrates profound inflammatory changes in NEC small intestine with increase in IL1β and TNFa producing macrophages, inflammatory signature in T cells with decreased suppressive signatures in Tregs accompanied by inflammatory changes in endothelial, epithelial, and fibroblast cells. We also identify a number of potential interactions such as MADCAM1-α4β7 between endothelial and T cells that could represent future therapeutic targets for NEC treatment. Our data provides a resource for future biomarker and therapeutic development in NEC.

## Methods

### Intestinal tissue acquisition and storage

Fresh small or large intestinal tissue from human neonatal and NEC samples were obtained from surgical resections in infants with IRB approval (S1 Table). The tissue was obtained from

the leading edge that is away from the site of intestinal injury. No consent was obtained for the samples as they were collected without any identifying information under a discarded specimen protocol that was deemed nonhuman research by the University of Pittsburgh IRB (IRB# PRO17070226). For single-cell sequencing and paraffin blocks, tissue was cryopreserved [58]. Briefly, intestinal tissue samples were cut into sub-centimeter pieces and cryopreserved in freezing media (10% dimethyl sulfoxide (DMSO) in fetal bovine serum (FBS)) in a slow cooling container (Mr. Frosty) at −80°C for 24 h, then transferred into liquid nitrogen for longterm storage. For paraffin blocks, tissue was fixed in 4% formalin for 48 h, transferred to ethanol until embedded in paraffin. The embedded paraffin blocks were stored until sectioned for analysis.

## Intestinal tissue digestion

Cryopreserved samples were processed as previously described [78]. Briefly, intestinal tissue samples were quickly thawed and washed in T-cell media that consists of: RPMI medium plus 10% FBS (Corning), 1X GlutaMax, 10 mM HEPES, 1X MEM NEAA, 1 mM sodium pyruvate (Gibco), 100 I.U/mL penicillin, and 100 μg/mL streptomycin. Next, intestinal tissue was incubated overnight in the same media with 1 μg/mL DNase and 100 μg/mL collagenase A. Tissue dissociation was performed on the gentleMACS Octo Dissociator with heaters (Miltenyi Biotec) using the heated human tumor protocol 1. Tissue was then filtered through a 70-μm nylon mesh cell strainer (Sigma) to make a single-cell suspension.

## Single-cell RNA sequencing

Single-cell suspension obtained from small intestine tissue digestion was washed in T-Cell Media (RPMI medium plus 10% FBS (Corning), 1X GlutaMax, 10 mM HEPES, 1X MEM NEAA, 1 mM sodium pyruvate, 100 I.U/mL penicillin, and 100 μg/mL streptomycin) twice followed by enrichment for live cells with the dead-cell removal kit using the MACS Cell separation system (Miltenyi kit # 130-090-101). Viable cells were counted with trypan blue using a hemocytometer. Using the MS or LS columns depending on the number of cells obtained, cells were separated into a CD45+ and CD45- fractions using the MACS Cells separation system with the CD45 bead enrichment (Miltenyi kit # 130-045-801). Cells were again counted on a hemocytometer with trypan blue. Libraries were made using the Chromium Next GEM Single-Cell 3′ kit v3.0 (10X Genomics) with a target of 5,000 cells, and 3′ GEM libraries were made separately, once from the CD45- fraction and once from the entire single-cell suspension from the tissue digest without any selection; 3′ GEM libraries were sequenced at Medgenome. Sequencing was performed on HighSeq lane with 2 samples per lane [79,80] or on an S4 NovaSeq lane.

## Bulk sequencing

RNA was extracted from snap frozen intestinal tissues samples using the Qiagen All Prep Kit (#80204). cDNA synthesis was prepped by the Yale Genomics core. RNA quality control (QC) was completed by MedGenome via Qubit Fluorometric Quantitation and TapeStation BioAnalyzer per company guidelines, with all samples passing the QC. Libraries were sequenced on the NovaSeq6000 for Paired End 150 base pairs for 40 million reads per sample. Paired-end reads fastq files were mapped with Hisat2.

## IMC staining and analysis

Biopsy-sized pieces of small intestinal tissue were fixed in formalin on day of collection, transferred to ethanol and subsequently embedded in paraffin in batches after 48 hours. Formalin-

fixed, paraffin embedded (FFPE) tissue was sectioned into 4 to 5 μm thick sections. Slides were deparaffinized using xylene and alcohol and placed in 1X antigen retrieval buffer (R&D Systems, #CTS013) at 95°C for 20 min. Next, slides were washed in distilled H20 (ddH20) and Dulbecco's Phosphate Buffered Solution (DPBS, Gibco). Tissue was blocked with 3% BSA in DPBS for 45 min at room temperature. Overnight incubation of antibodies (S5 Table) diluted in 0.5% BSA in DPBS at 4°C was performed. Slides were rinsed in DPBS with 0.1% TritonX100 twice and DPBS twice. Counterstain was performed with 100 μm LipoR-Ln115 and Ir-intercalator (1:1,000, Standard Biotools) in ddH20 at room temperature for 30 min. Slides were rinsed in ddH20 and then air dried.

## Selection of areas of interest

Regions of small intestinal tissue were selected manually to capture all layers of the intestine. The same sized area was scanned in all samples.

## IMC image acquisition

The Helios time-of-flight mass cytometer (CyTOF) coupled to a Hyperion Imaging System (Fluidigm) was used to acquire data. An optical image of each slide was acquired using Hyperion software and areas to ablate were selected as described above. Laser ablation was performed at a resolution of 4 μm and a frequency of 200 Hz. Data from the slides were acquired over 2 consecutive days in total of 18 image stacks from 9 samples (neonatal $n = 3$, NEC $n = 6$).

## IMC data segmentation and analysis

Data from Hyperion extracted as MCD and.txt files that were visualized using Histocat++ (Fluidigm). Further analysis of image data was performed using a recently published IMC segmentation pipeline [81] that was adapted to our dataset. Briefly, a Python script (https://github.com/BodenmillerGroup/imctools) was used to convert text format (.txt) files from data acquisition to tiff images. Spillover compensation was performed to minimize crosstalk between channels. The images where segmented in 2 steps. First ilastik [82], an interactive machine learning software was used to classify pixels as nucleus, cytoplasm, or background components. Training of the Random Forest Classifier was performed on $125 \times 125$ pixel sub stacks generated from original images using relevant markers (e.g., CD45, CD3, CD14, CD163, panCK, SMA). CellProfiler was used to identify nuclei, define cell borders, and generate cell masks and identify single-cell data from original images. Files and masks were loaded to histoCAT 1.7.6.1 [50], visualized and analyzed as follows. Images from Fig 5 were generated using the image visualization option in Histocat 1.7.6.1 [50]. Phenograph clustering of cells was performed using major lineage markers. IMC raw counts and median metal intensity for each cell for phosphoprotein markers were extracted as.csv for each cluster and data graphed using Graphpad (R) Prism 9. The sum of cells in each cluster was extracted from.csv files and used to calculate relative abundances. Individual expression data from each cell was exported for markers of interest and graphed using box and whisker plots in Graphpad (R) Prism 9.

Nearest neighborhood analysis was performed in histoCAT identifying neighboring cells within 4 pixels (1 pixel ~ 1 micron) to identify with interactions present in >10% of images with a $p$-value of 0.05 and 999 permutations. The program returns a matrix of cell–cell interactions that occur more frequently than random chance with the frequency of interactions expressed as a fraction of total images. There were 6 images for neonatal (2–400 × 400 microns sections scanned for each of the 3 samples) and 10 images for NEC (2–400 × 400 microns sections scanned for each of the 5 samples). Interactions within a row answer the question: is cell type X in the neighborhood of cell type 7 (or is Y surrounded by X). Those within a column

answer the question: is cell type 7 in the neighborhood of cell type X (x surrounded by 7). Interaction adjacency values are presented in S6 Table. Dot plots (Fig 6A) were produced by removing all adjacency interaction values involving the "other" cluster and sorting the differences between the interaction values in NEC and neonatal samples. The figure showed the top 20 adjacency interactions among the interactions with positive values in NEC, sorted from largest to smallest difference between NEC and neonatal samples (red dots) and the bottom 20 interactions with negative values in NEC, sorted from the smallest to the largest difference between NEC and neonatal samples (blue dots).

## TUNEL staining

Paraffin-embedded human intestinal tissue samples were sectioned, deparaffinized with xylenes (Sigma 534056-4L) and ethanol (Fisher BP2818-4), and fixed in 4% paraformaldehyde (PFA). Using the Click-iT Plus TUNEL Assay (Invitrogen C10619), samples were then permeabilized with Proteinase K solution. The TdT reaction was performed on the samples for 60 min at 37°C. Lastly, the Click-iT Plus reaction was performed on the samples for 30 min at 37°C.

## Immunofluorescence

Samples were blocked with 10% horse serum and incubated with primary antibodies (LYVE-1, 1:19, Biotechne AF2089). Samples were then incubated with secondary antibody (1:750, Invitrogen, A3216). Images were taken at 20× using Echo Revolve microscope. Fluorescence was quantified using ImageJ/Fiji. Equal size areas on immunofluorescent images were measured for integrated density and mean gray value. Mean gray values of background regions without fluorescence were also measured. Corrected total cell fluorescence (CTCF) was calculated from integrated density, area, and background fluorescent values.

## smFISH

smFISH was performed on frozen sections, with a modified smFISH protocol that was optimized for human intestinal tissues based on a protocol by Massalha and colleagues [83]. Intestinal tissues were fixed in 4% formaldehyde (FA, J.T. Baker, JT2106) in PBS for 1 to 2 h and subsequently agitated in 30% sucrose, 4% FA in PBS overnight at 4°C. Fixed tissues were embedded in OCT (Scigen, 4586), and 6 to 8 μm thick sections of fixed intestinal tissues were sectioned onto poly L-lysine-coated coverslips and fixed again in 4% FA in PBS for 15 min followed by 70% ethanol dehydration for 2 h in 4°C. Notably, unlike Massalha and colleagues [83], tissues were not permeabilized with PK prior to hybridization.

Tissues were rinsed with 2×SSC (Ambion AM9765). Tissues were incubated in wash buffer (20% Formamide Ambion AM9342, 2×SSC) for 30 to 60 min and mounted with the hybridization mix. Hybridization mix contained hybridization buffer (10% Dextran sulfate Sigma D8906, 30% Formamide, 1 mg/mL *E. coli* tRNA Sigma R1753, 2×SSC, 0.02% BSA Ambion AM2616, 2 mM Vanadyl-ribonucleoside complex NEB S1402S) mixed with 1:3,000 dilution of probes. Hybridization mix was incubated with tissues for overnight in a 30°C incubator. SmFISH probe libraries (S8 Table) were coupled to Cy5, TMR, or Alexa594. After the hybridization, tissues were washed with wash buffer for 15 min in 30°C, then incubated with wash buffer containing 50 ng/mL DAPI (Sigma, D9542) for 15 min in 30°C.

Tissues were transferred to GLOX buffer (0.4% Glucose, 1% Tris, and 10% SSC) until use. Probe libraries were designed using the Stellaris FISH Probe Designer Software (Biosearch Technologies, Petaluma, California, United States of America), see S8 Table.

## Imaging

smFISH imaging was performed on Nikon eclipse Ti2 inverted fluorescence microscopes equipped with 100× and 60× oil-immersion objectives and a Photometrics Prime 95B using the NIS element software AR 5.11.01. Image stacks were collected with a z spacing of 0.3 μm. Identification of positive cells was done using Fiji [84].

## Computational analysis

**Single-cell analysis.** The single-cell RNAseq data was processed using Cell Ranger 4.0.0 pipeline to align reads and generate a count matrix. Cell-free RNA, which often creates a background in scRNAseq was removed as follows: background cells were defined as cells with 100 to 300 UMIs and mitochondrial fraction below 50%. The average UMI counts for each gene in the background cells was subtracted from the respective counts in all other cells. Subsequently, cells with less than 200 expressed genes were filtered out and genes that were expressed in less than 3 cells were removed from data. Single-cell data was analyzed using R software version 4.0.2 and Seurat package (version 3.2.2 [85]).

All subjects were merged and cells with less than 1,900 UMIs, less than 1,000 genes per cell, or mitochondrial fraction above 30% were filtered out. In addition, cells with a fraction of erythrocyte markers ("HBG2," "HBA2," "HBA1," "HBB," "HBG1," "HBM," "AHSP," "HBZ") above 10% were filtered out. Cells were normalized and scaled using the SCTransform function (residuals of regularized negative binomial regression, log1p transformed and centered), with regression of sum of UMIs (vars.to.regress = "nCount_RNA"). Principal component analysis (PCA) was calculated based on the variable genes with the exception of mitochondrial ("^MT-") and ribosomal ("RP[LS]") genes. These genes were manually removed from the PCA, since they are prone to batch-related expression variability. Clusters were annotated based on a previous comprehensive cell atlas of the human small intestine by Elmentaite and colleagues [86] (see S2 Table for a complete list of markers obtained using FindAllMarkers in Seurat). Specifically, the summed expression of the top 10 markers for each of our atlas clusters were examined over the clusters of the atlas of Elmentaite and colleagues [86] and the published cluster name with highest expression was assigned.

Cell type structures (macrophages and dendritic cells, T cells, vascular endothelial and lymphatic endothelial cells, enterocytes and fibroblasts) were computationally extracted from the complete atlas based the cluster annotations. Cells were normalized and scaled using the SCTransform function (residuals of regularized negative binomial regression, log1p transformed and centered), with regression of sum of UMIs (vars.to.regress = "nCount_RNA"). PCA was calculated based on the variable genes with the exception of mitochondrial ("^MT-") and ribosomal ("RP [LS]") genes. Within the T cells, the Tregs+Th1 cluster was subset and reclustered, to enable proper splitting of the 2 cell populations. The resulting split cell annotations were presented in the main T cell UMAP. Re-clustering of fibroblasts subset revealed 2 clusters—fibroblasts and neurons.

DGE analysis used two-sided Wilcoxon rank-sum tests. DGE was only performed for genes with sum-normalized expression above $10^{-4}$ (or above $5 \times 10^{-5}$ in mid-bottom enterocytes). In addition, only genes that were expressed in 2 or more subjects and in at least 5 cells in each of the 2 subjects were retained. Q-values were computed using the Benjamini–Hochberg FDR correction [87]. GSEA included genes with sum-normalized expression above $10^{-4}$. The rnk file for GSEA included the sorted log2-fold expression between NEC and neonatal samples. In the Th1 T cell cluster, DGE analysis was not performed since neonatal cells were underrepresented. TLR4 signaling pathway genes were taken from REACTOME pathway database: "TLR6," "NOD1," "TAB1," "IRAK3," "TIRAP," "CHUK," "CREB1," "ATF2," "MAPK14,"

"TICAM1," "AGER," "DUSP3," "DUSP4," "DUSP6," "DUSP7," "ELK1," "TAB2," "FOS," "LY96," "PELI3," "TAB3," "TBK1," "HMGB1," "APP," "TICAM2," "IKBKB," "IRAK1," "IRAK2," "IRF3," "IRF7", "JUN," "PEDS-UBE2V1," "MEF2A," "MEF2C," "MAP3K1," "MYD88," "ATF1," "NFKB2," "NFKBIA," "NFKBIB," "IRAK4," "ECSIT," "PPP2CA," "PPP2CB," "PPP2R1A," "PPP2R1B," "PPP2R5D," "MAPK1," "MAPK3," "MAPK7," "MAPK8," "MAPK11," "MAPK9," "MAPK10," "MAP2K1," "MAP2K2," "MAP2K3," "MAP2K6," "MAP2K7," "PELI2," "PELI1," "SIGIRR," "RELA," "RPS6KA1," "RPS6KA2," "RPS6KA3," "RPS27A," "S100A12," "S100B," "SAA1," "NOD2," "MAP2K4," "MAP3K7," "BTK," "TLR1," "TLR2," "TLR3," "TLR4," "TRAF3," "TRAF6," "RPS27AP11," "UBA52," "RPS27AP11", "UBE2N," "MAPKAPK3," "IKBKG," "RIPK1," "RIPK2," "RPS6KA5," "MAP-KAPK2," "CD14," "IKBKE," "CDK1."

## Ligand–receptor analysis

Ligand–receptor analysis was performed following Martin and colleagues [88]. Briefly, this analysis computes for each pair of sender cell type A and receiver cell type B, and for each pair of ligand L and its matching receptor R, an interaction potential. The interaction potential is defined as the product of the mean expression of ligand L in cell A and the mean expression of receptor R in cell B. The interaction potentials are computed separately for the NEC cells and the neonatal cells, and their ratio is compared against a randomized dataset. In the randomized dataset, cells were re-assigned to the NEC and neonatal identities at random within each cell subtype, thus preserving the total number of NEC and neonatal cells. Random re-assignment was repeated 100 times and a probability that the observed ratio is higher than that expected by chance was computed using the normal distribution over the standardized ratio. The standardized ratio was defined as the difference between the real ratio and the average randomized ratio divided by the standard deviation among all randomized ratios. We only considered interactions that obeyed the following criteria: both ligand and receptor expressed in 10 or more cells in each of the respective cell types and in at least 10% of the cells in the cell type cluster in NEC or neonatal. Q-values were computed using the Benjamini–Hochberg FDR correction [87]. For each pair of cell types of interest, the top 25 interactions with q-value <0.01 (for some pairs less than 25 interactions passed this criteria) sorted by ratio were selected and plotted (S7 Table).

## Deconvolution analysis

Bulk data was filtered to include only coding genes (S9 Table), taken from Human_GRch38_91_ensemblBioMart and normalized to the sum of reads for each sample. Computational deconvolution was performed with cellanneal [89] (S10 Table). Only samples with Spearman correlations above 0.3 between the mixture data and the synthetic mixtures were considered. For the signature file, we included the mean expression per cell type for the original 8 cell type clusters shown in Fig 1B or for all cells, after breaking up cell type clusters into their subtypes (S11 Table). Since the 2 subsets of inflammatory macrophages showed a clear separation by disease status they were grouped together. For deconvolution of cells broken into subtypes, estimated proportions for each cell type were normalized internally over the respective cell subtype. P-values were calculated using two-sided Wilcoxon rank-sum tests. Q-values were computed using the Benjamini–Hochberg FDR correction [87] and were calculated over the internally normalized proportions of each respective cell subtypes.

## Cluster stability

Expression signatures for each cell type were calculated for all possible combinations of patients and were compared to the signatures based on the NEC portion of the full atlas using

Spearman correlations. Only genes with sum-normalized expression above $5 \times 10^{-6}$ were included in the correlation analysis. To examine the dependence of the correlations on the number of patients, we constructed 2 subsampled datasets: (1) datasets that include all possible combinations of $n < 6$ patients, where $n = 1, 2, 3, 4, 5$. We additionally included 3 bootstrap iterations for each patient combination. Values for the six-patient group were obtained by bootstrapping all cells. (2) For each set in (1) equally sized subsets of sampled cells from the complete atlas, while ignoring the patient information. Set (2) served as a control to assess the decrease in gene expression signature correlations that arose from the decrease in the number of cells sampled, rather than the number of patients.

### DNA extraction

DNA was extracted from snap frozen intestinal tissue samples with the Wizard Genomic DNA Purification Kit (Promega) per manufacturer's instructions for animal tissue.

### TCRβ Repertoire Library generation and analysis

Primers for various V and J gene segments in the *TRB* loci were used for amplification of rearranged CDR3B for each genomic DNA sample (ImmunoSeq TRB Survey Service, Adaptive Biotechnologies, Seattle, Washington, USA). Survey level of up to 500,000 reads/sample was used. Libraries were purified, pooled, and subjected to HTS using Illumina technology (Illumina, San Diego, California, USA) per manufacturer's protocol. ImmunoSeq software online was used to analyze clonality, percentage of T or B cells, sample overlap, percent productive template, CDR3B length, and VDJ use. Graphical representation of each repertoire is represented with hierarchical tree maps using available software (www.treemap.com).

### Code availability

All data generated in this study is available at the Zenodo repository under the following https://doi.org/10.5281/zenodo.5813397.

### Supporting information

**S1 Fig.** (**A, B**) Expression signatures of the NEC cell populations are stable with respect to number of cells and number of patients sampled. (**A**) Spearman correlations between the expression signatures of cell types between the full atlas and the atlas obtained by subsampling patients (black) or equal-sized groups of cells regardless of patients (red). Each subsampling contains all combinations of patients with additional 3 bootstrap iterations for each (Methods). Included are only cell types that were represented by 10 or more cells in at least 4 patients. (**B**) Median Spearman correlations of the complete atlas and the one obtained by sampling the indicated fractions of cells. Each value is a median over 20 bootstrap iterations. (**C**) Log10 scaled estimates of the proportion of enterocytes, fibroblasts, dendritic cells, macrophages, B cells, T-NK cells, enteroendocrine cells, and vascular/lymphatic endothelial cells based on computational deconvolution of the bulk RNAseq using the atlas single-cell populations (**Methods**). Each dot is a sample, fractions of enterocytes and fibroblasts normalized to the sum of cell fractions, remaining fractions normalized to the sum of all cells after excluding fibroblasts and enterocytes. Values are log10 (normalized data+$10^{-6}$); q-values are computed based on FDR correction for all cell populations in the full atlas (**Methods**). Gray lines are medians, black/pink boxes are 25–75 percentiles. Only samples with Spearman correlations >0.3 between the mixture data and the synthetic mixtures are shown (neonatal: $n = 4$, NEC: $n = 6$). The data underlying this figure is available at the Zenodo repository under the

following: https://doi.org/10.5281/zenodo.5813397 and in **S10 Table**.
(TIF)

**S2 Fig.** (**A**) Markers used to annotate the T-cell clusters in **Fig 3A and 3B**. Values are residuals of regularized negative binomial regression, log10 transformed, regressed to the sum of UMIs and centered. (**B**) Frequency of specific VDJ genes in neonatal and NEC samples (neonatal: $n = 7$, NEC: $n = 6$ from 7 neonatal and 4 NEC cases). (**C**) Complementarity determining region 3 (CDR3) length, number of deletions and insertions in NEC, and neonatal samples (neonatal: $n = 7$, NEC: $n = 6$ from 7 neonatal and 4 NEC cases). (**D**) Clonal representation of top clones in NEC and neonatal samples expressed as a proportion of the total clones in each sample. Black dots represents top 3 CMV clones, white dots are clones that were also notably present in intestine of human fetuses as published in [58], and gray dots are other clones (see Methods and **S4 Table**, neonatal: $n = 7$, NEC: $n = 6$ from 7 neonatal and 4 NEC cases). (**E**) Estimates of the proportions of T/NK cell populations based on computational deconvolution of the sequencing data. Each dot is a sample, proportions were renormalized over all T/NK cells, q-values are computed based on FDR correction for T/NK cells subsets only. Only samples with Spearman correlations >0.3 between the mixture data and the synthetic mixtures are shown (neonatal: $n = 4$, NEC: $n = 6$). The data underlying this figure is available at the Zenodo repository under the following: https://doi.org/10.5281/zenodo.5813397 and in **S10 Table**.
(TIF)

**S3 Fig.** (**A**) UMAPs colored by cell type and the expression of PDPN and CLDN5 in all cells in the atlas from **Fig 1B and 1C**. Vascular/lymphatic endothelial cells are marked by a red square. (**B**) UMAPs of re-clustered vascular and lymphatic endothelial cells colored by cell type and expression of PDPN and CLDN5. Cluster marked with a red arrow are a subset of inflammatory lymphatic endothelial cells that are PDPN negative. (**C**) UMAPs of re-clustered myeloid cells, colored by cell type, condition and the expression of TNF and IL1B. (**D**) UMAPs of re-clustered vascular and lymphatic endothelial cells, colored by condition and the expression of SELE and ICAM1. Color bar of expression plots is log10 (normalized expression). The data underlying this figure is available at the Zenodo repository under the following: https://doi.org/10.5281/zenodo.5813397.
(TIF)

**S4 Fig.** (**A**) Re-clustered atlas of the fibroblast cluster. (**B**) Cells annotated by condition. (**C**) Differential gene expression between NEC and neonatal cells for the fibroblast cluster only. Included are all genes with normalized expression above $10^{-4}$. Red dots are the top 30 most differentially expressed genes among the genes with q-value <0.01 and fold change above 2 or below 1/2. The data underlying this figure is available at the Zenodo repository under the following: https://doi.org/10.5281/zenodo.5813397 and in **S3 Table**.
(TIF)

**S5 Fig.** (**A**) Heatmap of normalized median expression of surface and phosphoprotein markers used to identify populations in IMC analysis (**Methods,** neonatal $n = 3$, NEC $n = 6$) depicted in B. (**B**) t-stochastic neighborhood embedding (tSNE) of IMC data clustered by RPhenograph and color coded by cluster (left-hand side) and condition (right-hand side). (**C–G**) Significantly elevated ligand–receptor interactions between cell populations explored. Shown are the log-ratios between the interaction potentials in NEC and neonatal samples (**Methods**). Maps include the 25 significant interactions (q-value <0.01) with highest fold change (**Methods**). In all interaction, maps sender population is on the y-axis, receiver population is on the x-axis. (C–G) Neo = Neonatal. The data underlying this figure is available at the Zenodo repository under the following: https://doi.org/10.5281/zenodo.5813397 and in **S6**

**and S7 Tables**.
(PDF)

**S1 Table. Samples used in the study. Demographics and allocation of samples used in the study.** *Samples sequenced but excluded from analysis for low correlation. For NGS: X indicates small intestine, + large intestine.
(XLSX)

**S2 Table. Markers of all cells and subset clusters. In the first tab: markers for the 8 cell type clusters, identified by the FindAllMarkers command in Seurat.** Markers included have a log-fold above 1 and expressed in at least 50% of the cluster cells. In all the rest of the tabs: Markers for each of cell subtype (Myeloid, T/NK cells, vascular/lymphatic endothelial cells, enterocytes, and fibroblasts) clusters (compared internally to the other clusters in the same cell subtype), identified by the FindAllMarkers command in Seurat. Markers included have a log-fold above 0.6 and expressed in at least 25% of the cluster cells.
(XLSX)

**S3 Table. Differential gene expression between NEC and neonatal for each cell type subset shown in the study.** Included are all genes with sum-normalized expression above $10^{-4}$ (macrophages, dendritic cells, T cells subsets, vascular endothelial cells, lymphatic endothelial cells, and fibroblasts) or $5 \times 10^{-5}$ (mid-bottom enterocytes). *P*-values were calculated using two-sided Wilcoxon rank-sum tests. Q-values were computed using the Benjamini–Hochberg false discovery rate correction.
(XLSX)

**S4 Table. TCRβ shared clones. Abundance, amino acid sequences, and antigen reactivity if known of the various clones in NEC and neonatal samples.**
(XLSX)

**S5 Table. IMC antibodies: antibodies used for imaging mass cytometry staining and analysis.**
(XLSX)

**S6 Table. IMC interactions: adjacency interaction values in NEC and neonatal samples.**
(XLSX)

**S7 Table. Ligand–receptor analysis table. Full table of ligand–receptor interaction analysis for all cell types.**
(XLSX)

**S8 Table. Probe library sequences. Sequences for the smFISH probe libraries used in this study.**
(XLSX)

**S9 Table. Mix table for deconvolution. Bulk RNA sequencing count table of all 14 subjects (4 neonatal, 10 NEC) before correlation filtration in deconvolution analysis.**
(XLSX)

**S10 Table. Deconvolution results. Proportions table from cellanneal analysis. Tab1 is deconvolution results using signatures of general 8 cell type clusters shown in Fig 1B. Tab2 is deconvolution results using signatures of cells split into subgroups of cell type.**
(XLSX)

**S11 Table. Signature table for deconvolution. Mean expression of major cell types from single-cell data.** Data in sum-normalized. Tab1 is mean expression of general 8 cell type clusters shown in Fig 1B. Tab2 is mean expression split into subgroups of cell type. (XLSX)

## Author Contributions

**Conceptualization:** Adi Egozi, Oluwabunmi Olaloye, Shalev Itzkovitz, Liza Konnikova.

**Data curation:** Lael Werner, Tatiana Silva, Blake McCourt, Richard W. Pierce, Xiaojing An, Fujing Wang, Kong Chen.

**Formal analysis:** Adi Egozi, Oluwabunmi Olaloye, Lael Werner, Richard W. Pierce, Xiaojing An, Dror Shouval, Shalev Itzkovitz, Liza Konnikova.

**Funding acquisition:** Shalev Itzkovitz, Liza Konnikova.

**Methodology:** Lael Werner, Tatiana Silva, Blake McCourt, Xiaojing An, Fujing Wang, Kong Chen, Jordan S. Pober.

**Software:** Adi Egozi.

**Supervision:** Kong Chen, Jordan S. Pober, Dror Shouval, Shalev Itzkovitz, Liza Konnikova.

**Validation:** Adi Egozi.

**Visualization:** Adi Egozi, Oluwabunmi Olaloye.

**Writing – original draft:** Adi Egozi, Oluwabunmi Olaloye.

**Writing – review & editing:** Shalev Itzkovitz, Liza Konnikova.

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
