## [Editor Report · Decision Letter 0]

10 Nov 2022

Dear Dr Konnikova, 

Thank you for submitting your manuscript entitled "Single cell atlas of the neonatal small intestine with necrotizing enterocolitis" for consideration as a Research Article by PLOS Biology.

Your manuscript has now been evaluated by the PLOS Biology editorial staff as well as by an academic editor with relevant expertise and I am writing to let you know that we would like to send your submission out for external peer review.

Once your full submission is complete, your paper will undergo a series of checks in preparation for peer review. After your manuscript has passed the checks it will be sent out for review. To provide the metadata for your submission, please Login to Editorial Manager (https://www.editorialmanager.com/pbiology) within two working days, i.e. by Nov 14 2022 11:59PM.

As a note: after some discussion within the team, we think your manuscript would be best suited for our Methods and Resources article type. Therefore, as you complete your submission we ask that you change the article type accordingly. For more information on Methods and Resrouces, see here: https://journals.plos.org/plosbiology/s/what-we-publish#loc-methods-and-resources-articles

Kind regards,

Luke

Lucas Smith, Ph.D.

Associate Editor

PLOS Biology

lsmith@plos.org

---

## [Decision Letter · Decision Letter 1]

4 Jan 2023

Dear Dr Konnikova,

Thank you for your patience while your manuscript "Single cell atlas of the neonatal small intestine with necrotizing enterocolitis" was peer-reviewed at PLOS Biology and I apologize again for our delay in sending you a decision. Your study has now been evaluated by the PLOS Biology editors, an Academic Editor with relevant expertise, and by several independent reviewers and I am writing to let you know that, in light of the reviews, we would like to invite you to revise the work to thoroughly address the reviewers' reports.

The reviews are appended below. As you will see, the reviewers appreciate the comprehensive and useful description of the changes associated with Necrotizing enterocolitis, provided here. However, they have raised a number of important concerns which would need to be thoroughly addressed before we can consider your manuscript for publication. Many of the reviewer critiques relate to the clarity and accuracy of the presentation, and the reviewers highlight the need to provide further explanation and justifications of the methods in order to strengthen this as a resource.

Given the extent of revision needed, we cannot make a decision about publication until we have seen the revised manuscript and your response to the reviewers' comments. Your revised manuscript is likely to be sent for further evaluation by all or a subset of the reviewers

**IMPORTANT - SUBMITTING YOUR REVISION**

*Re-submission Checklist*

*Published Peer Review*

*PLOS Data Policy*

*Blot and Gel Data Policy*

Sincerely,

Lucas

Lucas Smith, Ph.D.

Associate Editor

PLOS Biology

lsmith@plos.org

REVIEWS:

Reviewer #1: This is an elegant and detailed description of cellular changes associated with Necrotizing enterocolitis (NEC) a devastating neonatal gastrointestinal (GI) complication that is associated with significant morbidity and mortality.

The authors have examined this hard-to-study population, using state of the art methods that include the scRNA seq, next gen TCR seq with some protein-level validation using assays like TUNEL and smFISH where appropriate.

Specifically, scRNAseq was performed on 11 subjects (6 NEC and 5 neonatal samples). To enable a precise estimation of population proportions, computational deconvolution of bulk RNAseq (10 NEC and 4 neonatal samples) data based on the clusters identified in the scRNAseq dataset was performed. Further, next generation sequencing (NGS) of TCR� was performed to identify clonality changes associated with NEC (5 NEC and 9 neonatal samples). Finally, scRNAseq was combined with imaging mass

cytometry (7 NEC and 3 neonatal samples) data to define niche and cellular interactions enriched in NEC. 

Findings include numeric increase in various cell types in NEC including Macrophages. 

Further, qualitative Mac associated changes included upregulation of key inflammatory programs. Similar features also seen in DCs. T cell subset changes showed an increased frequency of TRBV10 and reduced TCRb length with the possibility of an increase in CMV-reactive T cell clones although the expression of IL17 was not detected in NEC. No significant change in Treg associated genes (rather a decrease in some of them) was noted. Other changes included numeric increase in lymphatic and vascular endothelial clusters with an upregulation of pro inflammatory programs and an increase in apoptosis and coagulation-related genes.

Epithelial cells changes were characterized by a reduction of villous-top APOA4+ cells and a relative increase in crypt-associated LGR5+ cells. Further, there was an increase in chemokines and Duox2+ expression by the epithelium. Increased interactions of endothelial cells with DCs and between hematopoietic and non-hematopoietic cell types was noted.

Overall, this is an impressive body of work using a number of high dimensional approaches. The main critique for such analyses is of course that this is descriptive. That said, it will serve as a useful and important resource for further studies into the pathogenesis and treatment of NEC. 

As mentioned by the authors, distinctions between medical therapy responders- and non-responders would be important next steps as would be further analyses of the pathophysiological pathways, especially characterizing the key pathogenic events that lead to adverse clinical outcomes

Another critique of the paper is regarding the writing style- while the flow is good and the manuscript is easy to follow, the discussion ends up being repetitive and inordinately long in places. As one example, the discussion about a4b7-directed Vedolizumab is ill-informed due to the relatively long latency of action of the drug that would preclude its use in these critically ill neonates. There are other areas where the discussion could be focused and streamlined into a few critical areas of pathogenic mechanisms and their significance. 

Finally, given that the authors are positioning this as a potential resource paper, it would be important to also provide detailed sc cluster tables (not just the deconvolution analyses).

Altogether, my enthusiasm for this paper is very high barring the caveats mentioned above. 

Reviewer #2: Major 

1. From Fig 1E, it would appear that the greatest change in celltype proportions in NEC is increased fibroblasts and decreased enterocytes. If we take these changes as pseudo-indicators of tissue injury, it indicates a variable level of tissue injury across the NEC samples in this dataset. If we assume that a 'normal' intestinal biopsy to be made up of mostly enterocytes as seen in the neonatal sample, NOT removing enterocytes and fibroblasts from 'all cells' would inevitably lead to observed changes in the proportions of other celltypes. In other words, if both neonatal and NEC samples had 100 total cells, ~90 of these cells in the neonatal group would ALWAYS be enterocytes and ~10 would be non-enterocytes. Whereas in the NEC group, the non-enterocyte cell number could be highly variable purely due to a change to enterocyte numbers. In my opinion, it would be more appropriate to show the proportional change of enterocytes and fibroblasts as a fraction of all cells first and then for the rest of the celltypes, look at the fractional change in the non-enterocyte/fibroblast fraction. This allows for a direct comparison between non-enterocyte proportions in neonatal and NEC samples.

2. Line 104: "This analysis revealed an increase in the numbers of..." Based on Fig 1E alone, it is not appropriate to say 'numbers' but rather proportion. Similar comment for Line 106: "...decrease in numbers..."

3. Line 118-119: It is not readily apparent how these genes are considered anti-inflammatory. The reference provided only potentially links TGFBI and anti-inflammatory macrophages.

4. Figure 2G: It would appear that only 3 NEC infants had increased macrophage proportions in Fig 1E, with all other infants having what appears to be nil macrophages. Yet there are now >3 NEC infant with increased inflammatory macs in Fig 2G. How is this possible? Is it just that in 1E there are overlapping datapoints in Fig 1E?

5. Figure S2A: It is not clear how the genes shown in Fig S2A define the ILC subset. It is almost identical to activated T cells? Please elaborate here and for all other subclustering analyses on how celltypes were labelled e.g. was it using signature genes from another study? Please reference.

6. Figure S2A: Color legend for average expression doesn't make sense given that the red color gradient starts from a negative value. How can average expression of the gene be a negative value?

7. Fig S2E: According to deconvolution, practically all of the bulk neonatal samples were naive T cells and yet in the sc-RNAseq in Fig 3B, naive T cells are the rarest T/NK/ILC cluster-type cell to be sampled? This does not make any sense as 3/4 neonatal control samples used in bulk RNA-seq was also used for sc-RNA-seq, suggesting they should be practically the same? Can the authors please comment on this discrepancy.

8. Figure 4L: It is not appropriate to perform statistical testing on repeated measurements sampled from 2 infants per group. Would suggest to remove 4L and state that J and K are representative images from n = 2 per group.

9. Figure 5H: There are very few crypt cells in neonatal samples. By combining crypt + mid-bottom villus into a single group, there is a potential bias that any differentially expressed genes may really just be identifying the fact that there are more crypt cells in NEC than neonatal group. For example, are LCN2, REG1A, REG1B and DMBT1 crypt-associated genes? Please verify that DEGs here are not due to crypt-cell bias in NEC group or only do DEG analysis on crypt OR mid-bottom villus separately.

10. Line 227: Please elaborate on what is included in the signature of TLR4 activation or the source of this signature.

11. Line 229-230: Given this statement and my suggestions on Figure 5H, it would be interesting to see if the increase to the TLR4 signaling pathway is still observed when comparing only the mid-bottom villus fractions between NEC and neonatal given the more balanced comparison.

12. Line 303-305: As far as I understood, the interaction analysis did not provide direct evidence of interaction between these celltypes but rather an indication that they may be interacting.

13. Line 308-313: Given the interest to the field of the role of Th polarization and its links to NEC pathogenesis, have the authors tried to determine if the activated T cells have a predominant Th signature of any kind? Or do they only detect some upregulation of Th17 genes but NOT that of Th1 and Th2?

14. Given the connection between using the word 'significant' has to being statistically significant, the authors need to more accurately describe their results when it is NOT statistically significant as trends. For example, Line 157-168, none of these are statistically significant.

Minor 

1. Figure 1: Visualization of 1E should be improved to be able to see datapoints clearly (for example, using more horizontal space where dots overlap or transparency or dot size). For example, in Fibroblasts NEC group there should be 10 datapoints but it looks like there is at most 6 dots there.

2. Figure S4B, color legend missing.

3. Supplementary Table 3: Is the 'Ratio' column referring to Log2 fold-change? Suggest to rename to be clear. If not, what does it mean to have a Ratio of 0.95? Is it 5% decreased in NEC vs neonatal?

4. Supplementary Table 5: There are several antibodies that are labelled with 'Harvard Core' for source. Please provide original source unless these antibodies are available only from Harvard Core.

5. Would recommend that n numbers are described in figure legends.

6. Line 200: There is no direct evidence shown in FigS3D-E that there is a correlative relationship between TNF/IL1B with SELE and ICAM1, e.g. the more TNF there is, the more SELE is present. The relationship is associative.

7. Line 203: Please elaborate on what LYVE-1+ cells are. What is the purpose of LYVE-1+ staining? Endothelial cell marker?

8. Line 240: First mention of the word IMC, unclear what that is.

9. Line 253: Unclear what a mf cell is.

10. Line 567-568: What does it mean that only edge piece was obtained for intestinal tissue acquisition? Is this a biopsy that contains both macroscopically NEC-afflicted inflamed tissue and non-inflamed tissue? More detail if possible should be provided for example, on the exact site of sample collection in NEC, i.e. Ileal, jejunal? 

11. Line 588, what is T-Cell Media? Please provide more details.

12. Line 601: What was quality controlled? I assume RNA quality, and what was the result? For example, was RIN over 8 for all samples?

13. Line 607: Fixed for how long? How was it processed before being embedded in paraffin or was it directly taken out of formalin into paraffin?

14. Line 610: Is this R&D systems?

15. Line 614: No company reference provided for LipoR-Ln115 and Ir-intercalator

16. Line 724: "Clusters were annotated based on markers from literature" but no references are indicated.

17. Line 728-730: Is this a repeat of Line 720-723?

18. There are a few instances of inconsistent unit labelling and likely exporting errors to watch out for. E.g. Line 581: ML, Line 609: uM, Line 199, Line 200

Reviewer #3: In this manuscript, combining scRNA-seq and other analyses, Egozi et al depicted a comprehensive atlas of the human gastrointestinal complication, necrotizing enterocolitis (NEC), laying a good foundation for further understanding of the disease. However, some issues need clarification for the publication in PLOS Biology.

1. Through the deconvolution method, the authors showed that the two kinds of myeloid cells, macrophages and dendritic cells are increased and decreased, respectively, in NEC (Fig. 1E). However, inflammatory genes were increased in macrophages and dendritic cells, and some reports that the authors referred to also revealed an enrichment of myeloid cells in NEC. Then, how to explain the decrease of dendritic cells and the increase of their inflammatory genes?

2. Some descriptions seem abrupt and lacked logicality. For example, the authors performed a search of the public database to determine the expanded clones in NEC, but they directly described the sequence known to bind to CMV without any explanations. 

3. In Fig. 4, the authors showed that there were more apoptotic endothelial cells in NEC, which is inconsistent with the increased number of endothelial cells as revealed in Fig. 1E. The apoptosis level is also in contrast to the high expression of pro-inflammatory signatures.

4. In Fig. 5D and 5E, the authors performed staining of villus top and bottom genes using normal tissues, and the NEC tissues should also be included to validate the loss of villus-tip enterocytes. 

5. The authors declaimed that this work help to understand NEC pathogenesis. However, the relative information is lacking. 

6. While inflammatory changes were found in NEC samples, are there any differences within NEC groups? Are there any relationships between gestational/postnatal age and inflammation severity?

7. ScRNA-seq data are sufficient to yield enough information about cluster change. What is the reason for choosing bulk seq-based computational deconvolution for this purpose? Do their two methods give different results?

8. There are some grammar and spelling mistakes, e.g, "This is correlates with" (line 60), "Intrestingly" (line 153), etc.

---

## [Editor Report · Decision Letter 2]

20 Mar 2023

Dear Dr Konnikova,

Thank you for your patience while we considered your revised manuscript "Single cell atlas of the neonatal small intestine with necrotizing enterocolitis" for publication as a Methods and Resources at PLOS Biology. This revised version of your manuscript has been evaluated by the PLOS Biology editors and by the Academic Editor. 

Our Academic Editor is largely satisfied by the changes made and feels the revision is quite thorough. However, the Academic Editor has one last technical request, which we think would need to be addressed before publication (request, below). Therefore, based on our Academic Editor's assessment of your revision, we are likely to accept this manuscript for publication, provided you satisfactorily address his/her remaining comment. We have also included a number of editorial and policy-related requests for you to address, below.

**EDITORIAL REQUESTS: 

1) COMMENT FROM ACADEMIC EDITOR: One technical point - in Fig R1 in the rebuttal letter, the authors provide both the linear and logarithmic scale graphs corresponding to a panel in Fig 1, but only have the linear scale in the actual paper. I'm guessing people who read this paper will have the same question as Reviewer #2 (see critique #4). I suggest they provide the logarithmic scale graph in the supplement or explicitly say that there are non-zero values in the text. 

2) TITLE: After some discussion within the team, we would like to suggest a minor edit to the title. If you agree, we suggest you change it to "Single cell atlas of the human neonatal small intestine affected by necrotizing enterocolitis"

3) ETHICS STATEMENT: Thank you for including an ethics statement about the use of human samples, here. For clarity, we would suggest the statement be edited slightly. If you agree this is accurate, we suggest you change statement to: 

"No consent was obtained for the samples as they were collected without any identifying information under a discarded specimen protocol, and that was deemed to be research that does not involve human patients, by the University of Pittsburgh IRB (IRB# PRO17070226)"

4) DATA REQUEST: Thank you for providing all the raw data for your study as a deposition on Zenodo. Looking through this dataset, I did not see the quantifications for the IHC (ex tunel staining, fig 4I). Apologies if I missed this, but if you have not provided this data, can you please add this either to Zenodo or as a supplemental table? Specifically, we ask that all individual quantitative observations that underlie this data be provided (we need each individual replicates AND the way in which the plotted; it should not present only the mean/average values).

--Please also ensure that figure legends in your manuscript include information on where the underlying data can be found, (for example, to each figure legend you can add the sentence, "the data underlying this figure is available at the Zenodo repository under the following https://doi.org/10.5281/zenodo.5813398"

We expect to receive your revised manuscript within two weeks. 

*Published Peer Review History*

*Press*

Sincerely,

Luke

Lucas Smith, Ph.D.

Associate Editor,

lsmith@plos.org,

PLOS Biology

---

## [Editor Report · Decision Letter 3]

13 Apr 2023

Dear Dr Konnikova,

Thank you for the submission of your revised Methods and Resources "Single cell atlas of the neonatal small intestine with necrotizing enterocolitis" for publication in PLOS Biology and for addressing our previous editorial requests. On behalf of my colleagues and the Academic Editor, Ken Cadwell, I am pleased to say that we can in principle accept your manuscript for publication, provided you address any remaining formatting and reporting issues. These will be detailed in an email you should receive within 2-3 business days from our colleagues in the journal operations team; no action is required from you until then. Please note that we will not be able to formally accept your manuscript and schedule it for publication until you have completed any requested changes.

As discussed over email - I have updated your supplemental figure 1 with the revised version that you provided me. Please do double check that everything looks in order to you. 

**IMPORTANT: I noticed one last minor issue regarding our previous editorial requests - please address the following point while addressing and formatting and reporting issues to come: 

1) Thank you for adding an updated ethics statement to your manuscript. Can you move the new statement into the methods section, and replace the previous statement?

Specifically, on line 597, "No consent was obtained for the samples as they were collected without any identifying information under a discarded specimen protocol that was deemed non-human research by the University of Pittsburgh IRB (IRB# PRO17070226)" should be updated to:

"No consent was obtained for the samples as they were collected without any identifying information under a discarded specimen protocol, and that was deemed to be research that does not involve human patients, by the University of Pittsburgh IRB (IRB# PRO17070226)."

PRESS

Sincerely, 

Lucas Smith, Ph.D.

Associate Editor

PLOS Biology

lsmith@plos.org